# Programmable DNA shell scaffolds for directional membrane budding

Michael T. Pinner[1,2] & Hendrik Dietz [1,2] ✉

In the pursuit of replicating biological processes at the nanoscale, controlling cellular membrane dynamics has emerged as a key area of interest. Here, we report a system mimicking virus assembly to control directional membrane budding. We employ three-dimensional DNA origami techniques to construct cholesterol-modified triangles that self-assemble into polyhedral shells on lipid vesicles, resulting in gradual curvature induction, bud formation, and spontaneous neck scission. Strategic positioning of cholesterols on the triangle surface provides control over the directionality of bud growth and yields daughter vesicles with DNA endo- or exoskeletons reminiscent of clathrin-coated vesicles. This process occurs with rapid kinetics and across various lipid compositions. When combined into a two-step process, nested bivesicular objects with DNA shells encapsulated between lipid vesicles could be produced. Our work replicates key aspects of natural endocytic and exocytic pathways, opening new avenues for exploring membrane mechanics and applications in targeted drug delivery and synthetic biology.

Membrane budding is a ubiquitous biological phenomenon in which parent lipid membranes release daughter vesicles for the transport of materials across the membrane. In biological systems, membrane curvature changes are often induced by curved proteins such as clathrin. With its distinctive triskelion shape, clathrin acts as a molecular scaffold by binding to membrane-bound receptor-adaptor protein complexes, followed by self-assembly into spherical cage-like structures that induce curvature in the cell membrane[1–3]. Recent studies confirmed clathrin's membrane-bending properties in vitro[4] and in cells[5]. We hypothesised that other materials capable of self-assembling into curved architectures beyond proteins could act as a molecular scaffold for membrane budding, with DNA origami emerging as a promising candidate. Herein, a several-kilobase-long, single-stranded 'scaffold' strand is mixed with an array of shorter oligonucleotides and exposed to a thermal annealing ramp to drive the folding of the DNA origami structure[6–8]. Previous efforts in the field related to lipid membranes have successfully tethered membrane-spanning DNA nanopores of various complexity to lipid bilayers using hydrophobic moieties such as cholesterol (chol)[9], ethyl-phosphorothioate[10] or tetraphenylporphyrin[11]. Notably, chol-labelled DNA origami objects tethered to lipid membranes retain

diffusive mobility influenced by factors like linker oligonucleotide length[12], paving the way for the 2D assembly of monomeric structures into multimeric lattices[13]. Prior studies have also shown that both curved[14–16] and planar[14] origami structures can induce tubulation in giant unilamellar lipid vesicles. On-membrane polymerisation of origami structures likewise resulted in the deformation of the underlying membrane, reminiscent of clathrin networks involved in cellular budding processes[13,17–21].

In this work, we describe an artificial, DNA-origami-based membrane budding system with spontaneous neck scission. At its core are three-dimensional membrane-interacting triangular subunits constructed from DNA with bevelled edges[22]. Strategically placed shape-complementary protrusions and recesses at the triangle edges direct the self-assembly of these triangular subunits into icosahedral shells through adhesive base pair stacking interactions[23]. Interactions with lipid membranes of giant vesicles (GVs) are promoted using chol moieties placed on the origami structure. The self-assembly of membrane-bound triangles induces local curvature and finally leads to the formation and eventual scission of buds without the need for active neck constriction by GTPases like dynamin[24]. The positioning of chol determines whether the bud grows inward or outward from the

[1]Laboratory for Biomolecular Nanotechnology, Department of Biosciences, School of Natural Sciences, Technical University of Munich, Garching bei München, Germany. [2]Munich Institute of Biomedical Engineering, Technical University of Munich, Garching bei München, Germany. ✉e-mail: dietz@tum.de

source vesicle, providing programmability for controlled membrane remodelling.

## Results

### DNA origami triangles as molecular scaffolds

We adapted a previously developed programmable icosahedral shell canvas, in which twenty equilateral DNA origami triangles form a closed icosahedron triggered by elevated concentrations of magnesium chloride for membrane-supported assembly (Supplementary Fig. 1 and Supplementary Tables 1–24)[22]. To provide the necessary hydrophobic interactions required for association with lipid membranes, we incorporated multiple cholesterol-bearing oligonucleotides (chol-oligos) at the shell-inward-facing surface of the constituent triangular DNA origami subunits (Fig. 1a and Supplementary Fig. 2). Single-stranded linker-handles protruding from the bottom face of the triangle served as attachment sites for chol-oligos (Supplementary Fig. 3a). We distributed them across the surface, keeping their relative position on each triangle side consistent and generally aiming for a symmetric layout. We hypothesised that the 2D diffusion of triangles on a fluid membrane can support the assembly of icosahedral shells

whilst pulling membrane material along, resulting in increasingly deformed membrane buds growing away from the parent membrane until the bud neck is cut and a DNA-shell-coated vesicle (DCV) is released into solution.

We screened chol density and positioning to balance the solubility of the triangles with membrane affinity (Supplementary Fig. 3b) and then validated their assembly into the expected icosahedral shells in the absence of membranes (Supplementary Fig. 3c). Introducing chol resulted in a shift from the previously observed relatively uniform icosahedral shells to a mixture containing both octahedral and icosahedral shells. We identified the proximity of linker handles to the outer edges and the chol configuration (proximal vs distal orientation) after hybridisation to a linker handle as influencing variables. By limiting the reach of each chol moiety (e.g. by increasing the separation of chols between neighbouring triangles in a shell assembly), we could largely revert the assembly to predominantly icosahedral structures and also reduce chol-mediated aggregation. We observed the fewest chol-mediated interactions using triangles with centrally-placed linker handles and proximal chol, and we have thus used this setup in all subsequent experiments.

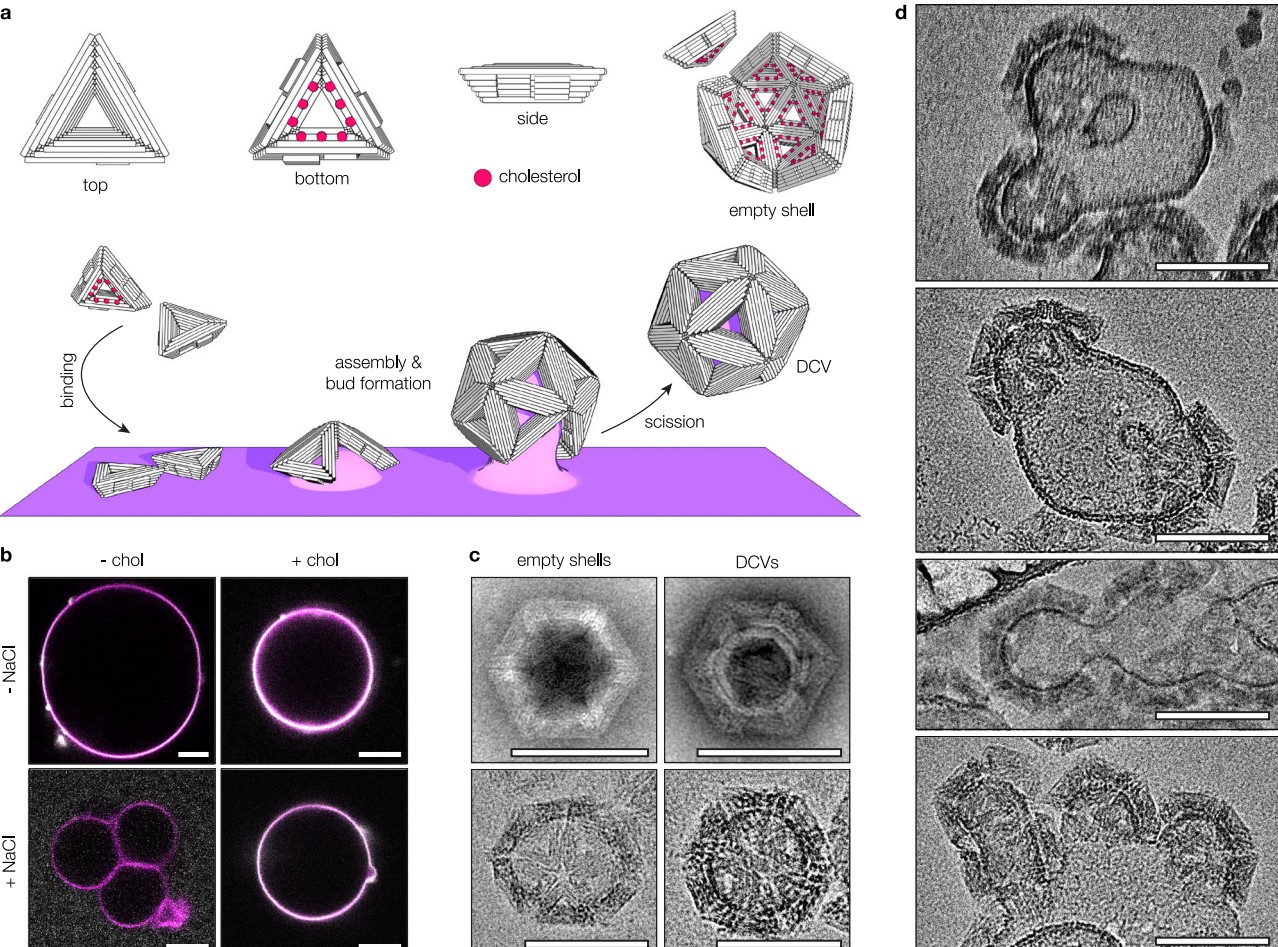

**Fig. 1 | DNA-shell-coated vesicles form by membrane budding. a** Schematic of a DNA origami triangle-based membrane budding system. The triangular structures are functionalised with chol moieties to tether them to the membrane of a lipid vesicle. A flat, rectangular segment is shown to represent a local portion of a giant vesicle (GV, purple). Upon increasing the MgCl₂ concentration, the triangles self-assemble into polyhedral shells, pulling the membrane underneath along to form a bud. Following spontaneous neck scission, a DNA-shell-coated vesicle (DCV) is released. **b** Pseudocoloured micrographs of DNA origami triangle attachment (grey) onto GVs (magenta). Triangles will adsorb onto GVs even if they are not decorated with chol (top left and top right panels). However, adding sodium

chloride suppresses this non-specific adsorption (bottom left panel), and membrane binding will only occur if chol is attached to the triangle (bottom right panel). Scale bars: 10 μm. **c** Transmission electron micrographs of empty DNA shells and DCVs. In negative stain transmission electron microscopy (TEM, top panels), lipid vesicles within DCVs appear as white rings inside the DNA shell. In cryogenic electron microscopy (cryoEM, bottom panels), the vesicle is visible as a dark ring inside the DNA shell. Scale bars: 100 nm. **d** CryoEM micrographs of large vesicle (LV) deformation upon assembly of membrane-bound DNA origami triangles. Bud curvature increases gradually and closely follows the intrinsic curvature of the assembling shells. Scale bars: 100 nm.

The addition of lipid membranes into the system introduced another variable, again leading to the occurrence of octahedral structures. This tendency varied depending on the amount of chol per triangle and the presence of lipid vesicles, detailed in Supplementary Figs. 4, 5 and 6. We believe this effect arises from a complex interplay of forces, including crowding of membrane-bound triangles, altered or biased relative alignment of triangles (possibly influenced by local curvature induction), and elastic deformation within the shells caused by less-than-ideal angles between the triangle subunits. Our shells share this structural ambiguity with clathrin, which likewise assembles into differently shaped cages without negatively impacting its role as a molecular scaffold, and we thus considered both shell types suitable for our objectives[3].

We prepared GVs composed entirely of DOPC and fluorescently labelled DOPE to serve as model membranes in our budding assays[25]. Aliquots of GVs and chol-decorated triangles were then mixed to anchor the nanostructures to the vesicles (Fig. 1b, detailed in Supplementary Fig. 7). The inclusion of NaCl in the reaction mixture minimised non-specific binding and ensured that the triangles attached to the vesicles in the correct orientation as defined by the positioning of their chol moieties[16]. We then triggered the assembly of membrane-bound triangles by increasing the $MgCl_2$ concentration to 60–65 mM and incubating the solution at 37 °C for up to 3 d. Compared to previous work where triangles were assembled at low NaCl concentrations[22], our setup requires significantly higher concentrations of $Mg^{2+}$ to outcompete sodium for assembly. Negative stain transmission electron microscopy (ns-TEM) imaging revealed abundant DCVs, characterised by dark DNA shells fully encapsulating bright lipid vesicles (Fig. 1c, top right). Occasionally, the shell-enclosed vesicles appeared deformed or cracked, which we attribute to staining and drying artefacts (Supplementary Fig. 8). To also study DCVs in their native state, we acquired micrographs of vitrified samples by cryogenic electron microscopy (cryoEM, Fig. 1c, bottom right; Supplementary Fig. 9a–c). This data generally agrees with the conclusions drawn from ns-TEM data, but the enclosed vesicles seen by cryoEM rarely had any defects, except for occasionally encapsulating even smaller vesicles themselves. In the cryoEM micrographs, the vesicle typically fills up the available space inside the origami shell, with the membrane in direct contact with the enclosed DNA shell.

As the size difference between DCVs and GVs spans multiple orders of magnitude, direct observation of DCV formation is challenging. To overcome this problem, we instead used large vesicles (LVs) of approximately 200 nm in our budding assays. Using isotonic $MgCl_2$ solution and shortening the time of shell assembly before vitrification, we could resolve intermediate states of DCV formation by cryoEM (Fig. 1d and Supplementary Fig. 9d). Whilst membrane patches free of DNA triangle subunits appear undisturbed, those with triangles on them had deformations that matched the curvature of the triangle assemblies (Supplementary Fig. 10). Notably, some vesicles carry multiple buds with clearly visible necks. This image data strongly supports the notion that the assembly of membrane-bound triangles induces curvature and that budding is the mechanism behind DCV formation.

## Morphology of DCVs

TEM analysis of budding reactions using triangles carrying either 3, 6, or 9 chols revealed a shift in DCV morphology with increasing cholesterol content (Fig. 2a and Supplementary Fig. 11). Higher numbers of chol moieties resulted in enhanced membrane binding, a greater proportion of octahedral DCVs, and an overall increase in DCV yield. For triangles with 6 or 9 chols, only a small fraction of empty shells was observed. In contrast, triangles carrying just 3 chol moieties frequently formed shells enclosing vesicles too small to fill the shell cavity (partial DCVs). Although some of these may result from staining artefacts, their enrichment at low chol content suggests weaker membrane binding

under these conditions. In addition to icosahedral and octahedral DCVs, a third population with pentagonal projections was observed across all variants. These likely represent partially assembled icosahedral shells—ranging from pentameric caps to half-icosahedral intermediates—that appear pentagonal in TEM projections.

We further analysed DCV formation efficiency by agarose gel electrophoresis using fluorescently labelled triangles and GVs (Fig. 2b). Lipid material colocalised almost exclusively with the higher-order shell assemblies and was virtually absent at the level of monomeric triangles and intermediates. Membrane-bound triangles that failed to form DCVs remained trapped in the gel pockets.

As with free triangles, efficient on-membrane assembly requires an optimal magnesium concentration (Supplementary Fig. 12). Magnesium modulates electrostatic repulsion and assembly kinetics: low $Mg^{2+}$ impairs shell formation, whereas excess $Mg^{2+}$ stabilises oligomeric intermediates and depletes available monomers. However, membrane-bound triangles appear less sensitive to excess magnesium, likely because their restricted lateral diffusion and the need to induce curvature reduce the formation and release of immature shell intermediates. DCVs also display moderately increased resistance to low-salt conditions, whereas free shells fall apart into monomers (Supplementary Fig. 13).

DCV formation efficiency is also affected by the ratio of triangles to vesicles, which depends on vesicle size and swelling conditions (Supplementary Fig. 14). Small vesicles (SVs) and LVs produced by extruding GVs showed a lower lipid optimum than untreated GVs, likely due to fewer multilamellar vesicles[26]. Similarly, budding from GVs formed in salt-free sucrose buffer showed a lower $Mg^{2+}$ optimum than from those swollen in caesium buffer, consistent with increased vesicle aggregation due to bivalent cations[27]. In both cases, the accessible membrane area appears to be the key factor. We also note that chol-free triangles, variants incapable of shell formation, and triangles added to GVs after several hours of membrane-free assembly all failed to produce DCVs efficiently (Supplementary Figs. 6b, 15 and 16). These findings underscore the critical role of on-membrane triangle assembly in DCV formation.

To evaluate whether the lipid vesicles were fully enclosed by well-ordered polyhedral DNA shells, we recorded single-axis tilt series to generate tomograms, allowing us to study DCVs in 3D (Fig. 2c and Supplementary Movie 1). The representative tomograms confirm full engulfment of the vesicle by closed shells featuring the symmetry properties expected by design. Occasionally, we found DCVs where the DNA origami shell had assembly 'scars' consisting of one to several missing triangles (Fig. 2c, d (arrows)). Considering that the increasingly high curvature at the bud neck can become a steric barrier to completion of the DNA origami shell, we believe the occurrence of scars to be a signature of the proposed budding process, where spontaneous membrane scission outpaces the completion of the origami shell. Supporting this, the pentagonal DCVs shown in Fig. 2a likely represent scarred icosahedral shells, in which triangles formed pentameric caps or half-shells.

## Biophysical parameters affecting budding efficiency

The addition of concentrated $MgCl_2$ solution as a trigger for shell assembly inevitably increases the osmolality of the sample buffer and may thus alter the biophysical properties of the membrane by deflating the vesicles. Indeed, hyperosmotic stress has previously been exploited to alter vesicle shape by membrane-bound DNA origami[16]. We studied the influence of membrane tension on DCV yield by mixing triangle-covered GVs with increasing amounts of glycine as an osmolyte. Interestingly, DCV yields correlated strongly with the osmotic environment. In the hypotonic regime, we observed moderate to low yields, with the lowest tonicity tested producing only a very faint band by agarose gel electrophoresis (Fig. 3a, Supplementary Fig. 17 and Supplementary Table 25). Conversely, as we transitioned into the

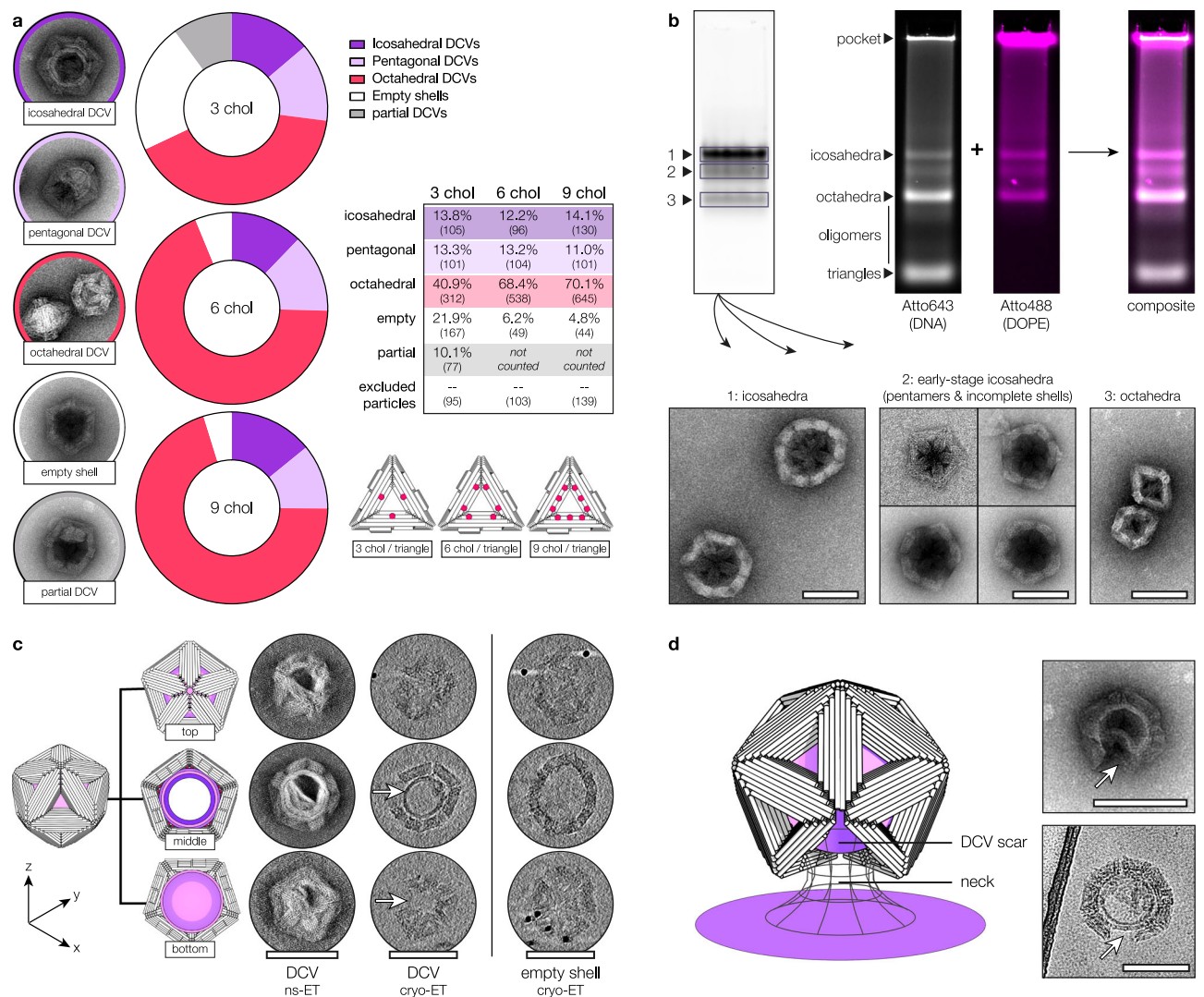

**Fig. 2 | Structural diversity of DNA-shell-coated vesicles. a** Relative abundance of particle subspecies as a function of cholesterol (chol) count. Besides octahedral and icosahedral DNA-shell-coated vesicles (DCVs), pentagonal DCVs–likely representing incomplete ('scarred') icosahedra–were frequently observed across all samples. Triangles with 3 chol/triangle yielded more empty shells and DCVs whose vesicles do not fill up the shell cavity ('partial DCVs'), suggesting weaker membrane binding. While some may be staining artefacts, their enrichment at low cholesterol supports a genuine trend. Particles with unclear morphology were excluded. Numbers in brackets indicate absolute particle counts (n). **b** Identification of shell subspecies by gel extraction and transmission electron microscopy (TEM). Electrophoretic mobility reflects particle size: icosahedral shells migrate slowest (top band), octahedral fastest (bottom), with intermediates (e.g. pentamers, half-shells) in between. Monomeric triangles run below shell assemblies but are often faint. DCV formation is assessed by agarose gel electrophoresis using fluorescently labelled DNA or ethidium bromide (white) and a small fraction of labelled DOPE lipids in the vesicle mixture (magenta). Lipid signal colocalises only with assembled shells, indicating membrane material migrates primarily as part of DCVs. DNA retained in the gel pocket reflects vesicle-bound triangles; the presence of a monomer band indicates saturated membranes. Pockets were oversaturated to visualise faint bands. **c** Summed z-slices depicting the top, middle and bottom segments of DCVs and an empty shell obtained by electron tomography (ET) of negatively-stained (ns) or vitrified (cryo) samples. The lipid vesicle is visible in the middle segments of DCVs, confirming engulfment. In the vitrified example, a gap in the DNA shell (arrows) also exposes the vesicle in the bottom segment. Cryo-images were Gaussian-blurred to enhance contrast. **d** DCVs occasionally appear incomplete with 'scars' of missing triangles (arrows). Steric hindrance at the bud neck can prevent closure, leaving gaps in the shell that remain visible after neck scission. Top: ns-TEM; Bottom: cryoEM. Scale bars: 100 nm. Source data are provided as a Source Data file.

hypertonic regime, we noted a marked and progressive increase in DCV yields, with the most pronounced yield achieved at the highest buffer osmolality. Although the tonicity screen used lower NaCl concentrations than typically needed to prevent non-specific adsorption, DCV yields in the mildly hypertonic range were comparable to those of the control prepared at standard conditions, suggesting minimal interference with budding. When magnesium levels were not increased to trigger assembly, we also did not observe directed budding as for DCVs in both isotonic and hypertonic buffers, further confirming the necessity for assembly (Supplementary Fig. 18). These findings reveal a direct relationship between membrane tension and DCV yields,

reinforcing the notion that DCVs emerge from intricate membrane remodelling processes.

Our system's sensitivity to membrane tension draws a parallel to clathrin-mediated endocytosis (CME) which depends on actin engagement for budding under increased membrane tension[28]. Another point of similarity is that the assembly of both our DNA triangles and clathrin itself is sufficient to deform model membranes in vitro[4]. However, vesicle deformation has also been demonstrated by attaching DNA origami structures to lipid vesicles using chol as an anchor, not requiring the assembly of subunits[14]. Having demonstrated the necessity of triangle assembly for DCV formation (Supplementary

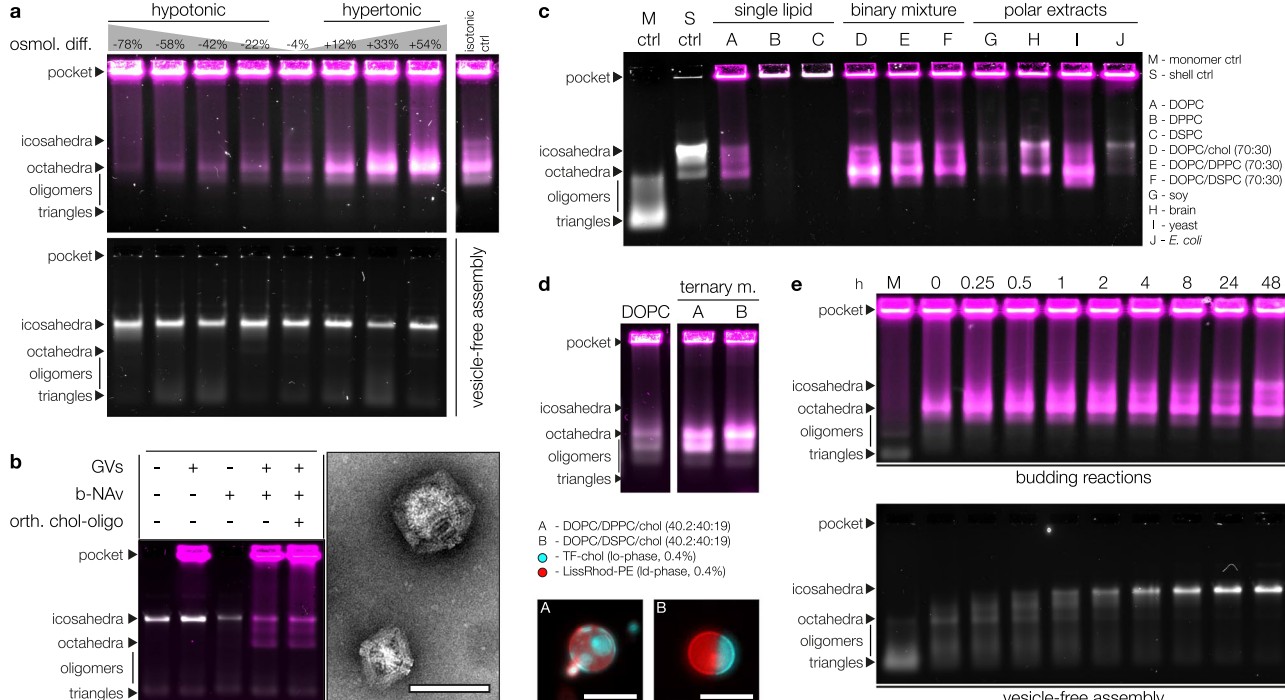

**Fig. 3 | Biophysical characterisation of DNA-shell-coated vesicles. a** Agarose gel of DNA-shell-coated vesicles (DCVs) prepared under varying osmotic conditions. Samples in the top gel contain triangles (grey, EtBr) and giant vesicles (GVs, magenta, DOPE-Atto643), whereas the bottom gel contains triangles only and serves as assembly control. Glycine and MgCl₂ were added at increasing concentrations to vary sample tonicity and promote triangle assembly. Budding efficiency strongly correlates with tonicity, working best under hypertonic conditions and coming to a near halt at the most hypotonic condition tested. **b** Left: Agarose gel of DCVs prepared by using biotin-NeutrAvidin (b-NAv) interactions to connect origami triangles (grey, Atto643) and GVs (magenta, DOPE-Atto488). The addition of orthogonal chol-oligos did not influence DCV yields. Right: TEM micrographs of DCVs obtained by this approach. Scale bar: 100 nm. **c** Agarose gel of DCVs obtained from GVs of varying lipid composition. Only GVs composed entirely of high-melting lipids prevented DCV formation. Grey: DNA (EtBr); Magenta: Lipids (DOPE-Atto643); **d** Top: Agarose gel comparing DCVs obtained from GVs composed of DOPC or phase-separated ternary lipid mixtures. Phase-separated vesicles yielded more DCVs, suggesting an influence of phase boundaries on the budding mechanism. Grey: DNA (Atto643); Magenta: Lipids (LissRhod-PE); Bottom: Micrographs of exemplary phase-separated vesicles showing the lipid-ordered (cyan, TopFluor-chol) and disordered (red, LissRhod-PE) phases. Scale bars: 20 μm. **e** DCV formation kinetics (top) compared against vesicle-free assembly kinetics. The lipid-channel image has been despeckled for clarity. Grey: DNA (EtBr); Magenta: Lipids (DOPE-Atto643); Source data are provided as a Source Data file.

Fig. 15), we next investigated the roles of chol insertion and outer leaflet expansion in inducing curvature and budding. To evaluate this, we prepared GVs using a lipid mixture containing 3% biotinylated DOPE and replaced chol-oligos with biotinylated oligonucleotides linked to NeutrAvidin (b-NAv) as membrane anchors. This allowed DNA triangles to attach to the membrane via NeutrAvidin-biotin interactions without inserting foreign lipids into the outer leaflet. Remarkably, this approach still resulted in DCV formation, and adding excess chol-oligos with an orthogonal sequence not complementary to the triangle linker handles did not improve yields (Fig. 3b and Supplementary Fig. 19). Compared to DCVs obtained using chol, the distribution between icosahedral and octahedral DCVs is more balanced when using b-NAv, though overall DCV yields appear to be slightly lower. We hypothesise that a sufficiently high density of NAv on the membrane may act as a protective barrier between both components, increasing their distance from one another and thereby altering electrostatic influences. These findings show that chol insertion is not required for DCV formation, supporting the idea that membrane buds are primarily shaped by shell assembly.

In our study, we primarily used vesicles composed of DOPC, supplemented with a small fraction of fluorescently labelled DOPE species. The high fluidity and homogeneity of DOPC membranes provide idealised conditions that do not necessarily reflect the properties of biological membranes. We next investigated the versatility of our budding system using GVs with diverse lipid compositions. These included single-lipid species compositions

(low-melting DOPC, and high-melting DPPC and DSPC), binary mixtures (70% DOPC with either 30% chol, DPPC, or DSPC), and four polar extracts of natural origin (derived from soy, brain, yeast, and *E. coli*). With the exception of GVs composed entirely of high-melting lipids, DCVs successfully formed from all tested compositions with varying efficiencies (Fig. 3c and Supplementary Fig. 20). The successful formation of DCVs from binary mixtures containing high-melting lipids suggests that the overall membrane fluidity, rather than the mere presence of high-melting lipids, is a critical factor in determining budding success. This is supported by the fact that low-melting DOPC constituted the majority (70%) of these mixtures, rendering the bilayer overall fluid.

Another important factor for DCV formation is neck scission. Constriction of the bud neck may be driven by spontaneous and induced curvature at the neck due to a high triangle density on the membrane and adhesion forces[29,30]. Scission can also be helped by line tension in phase-separated membranes, where the phase boundary acts as a linear defect along which fission is facilitated[31]. Indeed, when comparing DCV yields from pure DOPC membranes versus phase-separated membranes composed of DOPC/DPPC/chol or DOPC/DSPC/chol (40:40:20 mol%), the latter generally yielded more DCVs (Fig. 3d and Supplementary Fig. 21). Consistent with previous reports[32], we also observed preferential binding of DNA nanostructures to the ld-phase at low magnesium concentrations, with a shift towards the lo-phase upon triggering assembly (Supplementary Fig. 22).

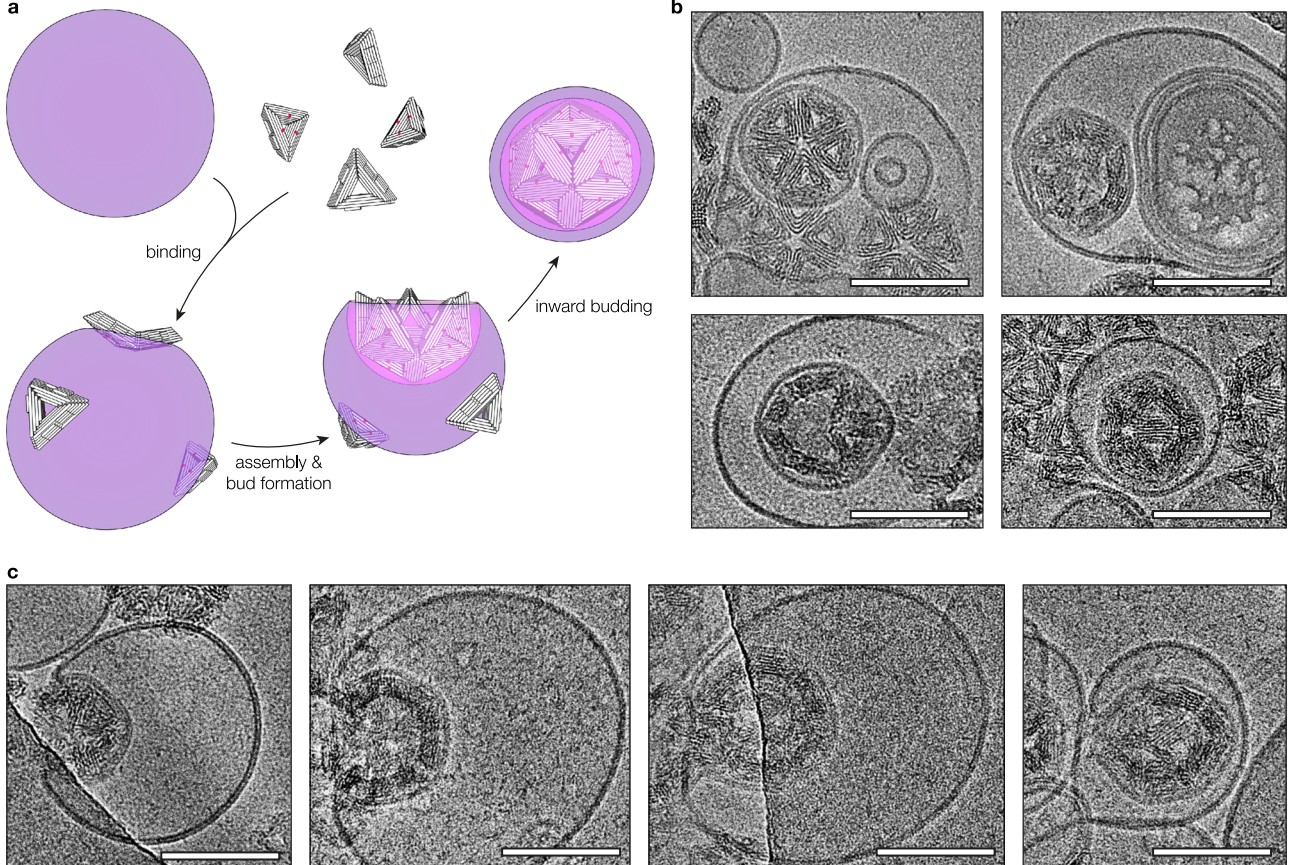

**Fig. 4 | Vesicle-coated DNA shells. a** Illustration of vesicle-coated DNA shell (VCD) formation by budding into its parent vesicle. The budding direction is reversed with respect to DCVs by shifting the linker handles for chol-oligos from the shell-inner face to the shell-outer face of the triangle subunits. **b** Representative cryoEM micrographs of VCDs within their parent vesicles. **c** Representative cryoEM micrographs of inward-oriented membrane buds at different stages. All scale bars: 100 nm.

We next analysed the kinetics of DCV formation by starting the assembly reaction and collecting aliquots at various time points. The budding process was rapid, with a substantial number of DCVs already present at the 0-min time point (Fig. 3e and Supplementary Fig. 23). We attribute the rapid kinetics to the membrane confinement of the origami triangles, which generates elevated local concentrations and restricts mobility to two dimensions. Consistent with earlier observations, the majority of DCVs formed initially exhibited an octahedral geometry. Over time, however, a growing fraction of icosahedral DCVs was detected. This progression mirrors the behaviour observed in vesicle-free control assemblies, where DNA origami triangles rapidly assemble into intermediates resembling octahedra in the agarose gel, which subsequently mature into icosahedral shells. However, the maturation of octahedral DCVs into icosahedral ones appears restricted. We believe that size constraints imposed by the vesicle inside the DCV stabilise its configuration and limit the expansion of the inner cavity necessary to form larger icosahedral species.

## Inward budding

Having studied the formation and characterisation of DCVs, we next investigated the directional reversal of this process. DCV budding mimics endocytic processes with the important difference that in our system, the triangular subunits approach the parent vesicle from the exterior. By shifting the placement of chol on the triangle surface from the bottom (shell-inward) to the top (shell-outward) face, we hypothesised that we could reverse the budding directionality in a process reminiscent of exocytosis. As the triangles would still bind to the

vesicles from the outside, we expect the formation and release of vesicle-coated DNA shells (VCDs) into the lumen of the parent vesicle (Fig. 4a), effectively creating an endosome-like compartment with a DNA origami endoskeleton. We prepared LVs by extruding GVs through 200 nm-sized pore filters to facilitate cryoEM imaging, mixed them with triangles carrying one chol moiety per side and triggered shell assembly. Direct imaging with cryoEM (Fig. 4b, Supplementary Fig. 24) revealed VCDs as lipid vesicles tightly wrapped around icosahedral shells located within their respective parent vesicles. Occasionally, we also observed free VCDs, presumably set free by the bursting of the parent vesicle, and the presence of holes in the DNA shell similar to those observed in DCVs.

To capture the dynamics of the suspected budding process, we prepared vitrified samples with shortened assembly times and analysed them by cryoEM. The micrographs we obtained revealed the gradual deformation of the parent vesicle in response to the assembly of membrane-bound triangles, leading to the formation of inward-growing buds (Fig. 4c). In contrast to fully formed VCDs, these particles remain connected to the parent membrane, sometimes just by a narrow bud neck.

We finally explored the creation of bivesicular shell structures by combining outward and inward budding reactions in a two-step process using triangles with sequence-orthogonal linker handles on both faces (Fig. 5a). First, we produced DCVs by hybridising chol-oligos to the triangles' bottom face and mixing the triangles with GVs as described earlier. After isolating the DCVs by centrifugation, we prepared LVs and coated them with chol-oligos complementary to the linkers on the DCV's outer face. Mixing the DCV concentrate with

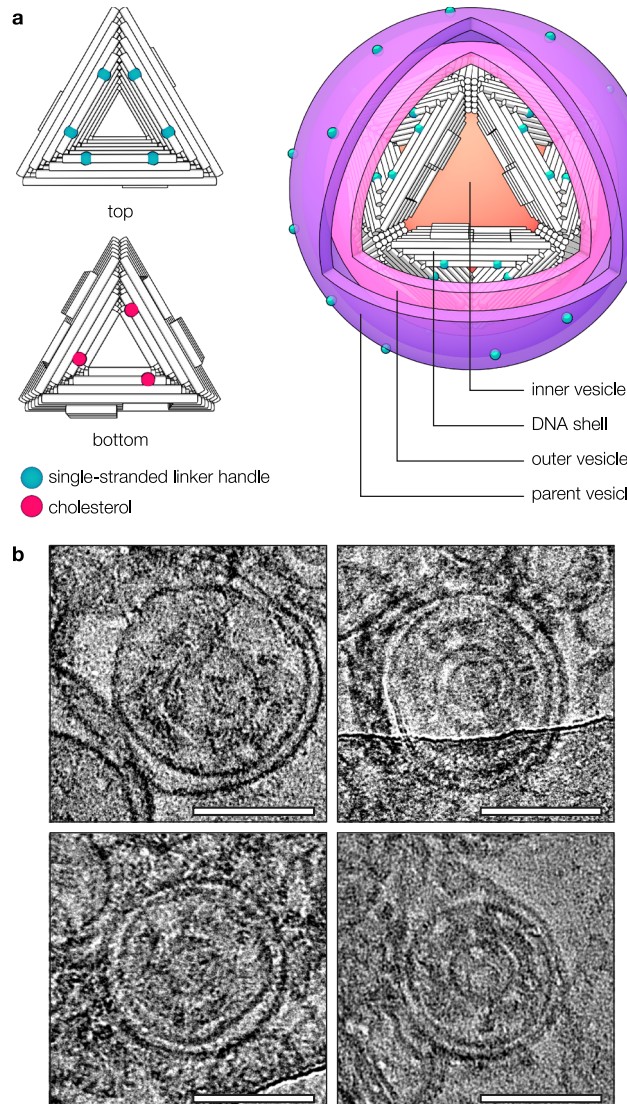

**a**

top

bottom

● single-stranded linker handle
● cholesterol

inner vesicle
DNA shell
outer vesicle
parent vesicle

**b**

**Fig. 5 | Multivesicular DNA shells. a** Illustration of a bivesicular DNA shell (vesicle-coated DCV; VCDCV) contained within its parent vesicle, produced in a 2-step budding assay using triangles with orthogonal linker handles on their top and bottom faces. In the first step, the bottom face linkers are hybridised to chol-oligos for DCV formation by outward budding. Then, DCVs are mixed with LVs carrying single-stranded linker handles complementary to those on the shell-outer face of the triangles, resulting in inward budding and VCDCV formation. As a result, VCDCVs have four layers: An inner vesicle stemming from the first step (orange), the DNA origami shell (white), an outer vesicle (pink) coupled to the shell by complementary linker handles, and the parent vesicle (purple) with its DNA linker handles (green blobs). **b** cryoEM micrographs of VCDCVs. Images were Gaussian blurred and levelled for clarity. Scale bars: 100 nm.

these LVs and incubating at 37 °C resulted in the formation of vesicle-coated DCVs (VCDCVs), as confirmed by cryoEM (Fig. 5b, Supplementary Fig. 25). Like VCDs, VCDCVs were typically enclosed within their parent vesicles as a result of the proposed budding mechanism. These particles may be considered simple examples of synthetically created organelles featuring a cytoplasm-like compartment (the inner vesicle) and a periplasm-like compartment (the void between the outer envelope membrane and the inner lipid membrane). The icosahedral DNA shell acts as a stabilising mechanical skeleton, offering engineering options for including additional molecular functionalities.

## Discussion

In a related study, Zhan et al. extracted membrane material from donor vesicles using chol-decorated DNA origami structures, but mechanistic details regarding the underlying process remain to be elucidated[20]. Our work complements this study by demonstrating the universality of the molecular scaffolding concept and identifying budding as the underlying process behind DCV formation. We developed a DNA origami-based membrane budding system that recapitulates key aspects of CME without relying on components of cellular budding machineries. Like clathrin, the DNA origami triangles function as molecular scaffolds whose self-assembly into cage-like structures induces membrane deformation and ultimately drives the formation of vesicles. Free energy is supplied by the self-assembly process and relayed onto the bilayer by either chol or biotin-NeutrAvidin linkages, mimicking the way clathrin associates with membranes via adaptor proteins. Using average stacking energies reported for DNA nanostructures ($-2.97\,k_BT$, $T = 300\,K$), we estimate the free energy of icosahedral assembly (30 edges with 16 base-stacking interactions each) at approx. $-1400\,k_BT$, and $-570\,k_BT$ for octahedra[33]. Our estimates exceed the energetic cost of bending DOPC membranes into spherical vesicles ($460\,k_BT$), derived from Helfrich's theory ($E = 8\pi k_C$, where $k_C = 18.3\,k_BT$ at 298 K), and support our experimental observations[34,35].

Previous studies have shown that high surface coverage of membrane-bound DNA origami is often required for vesicle deformation[14,16]. In contrast, our system induces budding even at low surface densities (Fig. 1d), likely due to the cooperative assembly of DNA triangles. This modular process allows local enrichment and gradual membrane bending driven by the free energy of assembly without the need for global coverage. Indeed, DCV formation occurs across a wide range of origami-to-vesicle ratios (Supplementary Fig. 14), indicating that membrane remodelling is governed more by local assembly dynamics and diffusion-driven encounters than by overall density. When vesicle numbers are too low, yield is limited by the available membrane area. Conversely, an excess of vesicles dilutes the local concentration of triangles per vesicle, reducing the likelihood of forming sufficiently large assemblies to induce the curvature required for budding.

Both clathrin cages and DNA origami shells demonstrate structural flexibility, forming various geometries while maintaining their scaffolding function[3]. When assembled on lipid membranes, the DNA triangles shifted notably from forming icosahedral to octahedral shells, influenced by factors such as lipid composition and the type of membrane tether. Similar polymorphic behaviour has been reported for DNA tubules composed of triangular subunits, where thermal fluctuations during loop closure were proposed as a cause, and increased design complexity as a potential solution[36,37]. However, in free assembly reactions, the vast majority of shells formed are icosahedral, suggesting membrane properties such as bending rigidity as major influencing variables governing both shell geometry and budding efficiency. The system may be optimised to yield more icosahedral DCVs by screening membranes of varying rigidities or increasing the complexity of the shell system, although both alterations could negatively impact DCV yield. We also suspect that electrostatic interactions between the membrane and the DNA origami, as well as molecular crowding effects, play a role. However, a detailed analysis lies beyond the scope of this study.

The scars observed on some DCVs point towards steric hindrance at the bud neck, preventing shell closure. While these defects could, in principle, be repaired by incorporating free triangles, the efficiency of this depends on both their availability and their assembly state. Here, the assembly of free triangles at elevated $Mg^{2+}$ concentrations may result in oligomers that are too large to fit into the scars of most DCVs. However, the rapid kinetics of DCV budding may outpace free triangle

oligomerisation, allowing their incorporation into the DCVs and explaining why only a subset exhibits scars.

Our system demonstrates robust performance across a variety of lipid compositions, including natural lipid extracts, with rapid kinetics highlighting its versatility and potential for broader applications. Nonetheless, we identified limitations under specific conditions. DCV formation was impaired in membranes composed entirely of high-melting lipids (DPPC, DSPC), and yields were reduced under hypotonic conditions. These constraints can be explained by the Helfrich Hamiltonian, which relates the energetic cost of membrane deformation to the mechanical properties of the bilayer[34]. Higher bending rigidities of gel-phase membranes and increased membrane tension under hypotonic conditions impose energetic penalties directly impacting DCV yields[38]. While clathrin underlies the same limitations, biological membranes make use of the cytoskeleton to enable CME even under high membrane tension[28]. Future work could explore active neck constriction to overcome the current limitations of our system.

In summary, we present a versatile platform for engineering vesicles of controlled size featuring addressable endo- or exoskeletons. This approach can be extended to encapsulate molecules of interest during the budding process, potentially leading to molecular payload applications. Using nested bivesicular structures, it becomes possible to design scenarios where two different molecular payloads are delivered simultaneously but separately. This work may thus pave the way for innovative applications in targeted drug delivery, vaccine development, and synthetic biology. The potential to manipulate membrane dynamics at the nanoscale also opens new horizons in synthesising artificial cells and organelles, offering a valuable tool for researchers in their quest to harness the power of nanotechnology for practical and therapeutic purposes.

# Methods

## Materials
CsCl (≥99.999%, cat. #8627.1), solid MgCl₂ (≥99%, cat. #HN03.3), Sucrose (≥99.5% p.a., cat. #4621.1), Glycine (≥99%, cat. #0079.2), Ficoll400 (cat. #CN90.3) and OrangeG (cat. #0318.2) were purchased from Carl Roth. NaCl (≥99%, cat. #746398; 5 M solution, cat. #S5150), Glucose (≥99.5%, cat. #RDD016), Chloroform (≥99.5%, cat. #C2432), Triton-X100 (cat. #X100), and BSA (cat. #A4919) were purchased from Sigma Aldrich. 1 M MgCl2 solution was purchased from Honeywell Fluka (cat. #63020). Agarose was purchased from Invitrogen (cat. #16500). 1,2-dioleoyl-sn-glycero-3-phosphocholine (DOPC, >99%, cat. #850375), 1,2-dipalmitoyl-sn-glycero-3-phosphocholine (DPPC, >99%, cat. #850355), 1,2-distearoyl-sn-glycero-3-phosphocholine (DSPC, >99%, cat. #850365), 1,2-dioleoyl-sn-glycero-3-phosphoethanolamine-N-(cap biotinyl) (biotinylated DOPE, >99%, cat. #870273), polar lipid extracts from soy, brain, yeast, and *E. coli* (cat. #541602, 141101, 190001, 100600), TopFluor cholesterol (>99%, cat. #810255) and Lissamine Rhodamine B-DOPE (>99%, cat. #810150) were obtained from Avanti Research. Atto488- and Atto643-modified DOPE (cat. #AD 488-161, AD 643-161) were obtained from Atto-Tec. NeutrAvidin was purchased from Thermo Scientific (cat. #31050). Regular DNA oligonucleotides were purchased from IDT, and modified oligonucleotides were obtained from Biomers. Scaffold DNA was produced biotechnologically in-house. Buffer osmolalities were measured using an Osmomat 3000 freeze point osmometer (Gonotec). For buffer compositions, refer to Supplementary Table 26.

## DNA origami folding, purification and quantification
DNA origami triangles were folded as described previously[22]. Briefly, circular scaffold strands of bacteriophage origin (sc8064, 50 nM; Supplementary Table 1) and staple oligonucleotides (200 nM per staple; Supplementary Tables 2–23) were mixed in folding buffer and exposed to a thermal annealing ramp (65 °C, 15'; 58–53 °C, 1 h/°C; stored at 20 °C) in a Tetrad2 (Bio-Rad) thermocycler. Origami structures were designed and edited using caDNAno 2 (https://cadnano.org/)[39]. For some experiments, structures were fluorescently labelled by including Atto643-labelled oligonucleotides in the folding reaction. The folded product was purified by agarose gel electrophoresis by excising the leading band, crushing the gel pieces, and centrifuging them in 0.45 μm Costar Spin-X spin columns (Corning) for 10 min at 8000 x g. When needed, the purified structures were washed and concentrated by ultrafiltration (Amicon Ultra 0.5 mL, 100 kDa cut-off, Millipore), exchanging the buffer for sodium buffer. Origami solutions were quantified using a NanoDrop8000 spectrophotometer (Thermo Scientific).

## Vesicle preparation, processing and quantification
Giant vesicles were prepared by gentle swelling of dry lipid films on polyvinyl alcohol gels as described previously[25], with slight modifications. A petri dish (⌀ 10 cm) was plasma cleaned in a glow discharge device, and then 1800 μl of a 5% solution of polyvinyl alcohol (Mowiol 28–99, SigmaAldrich, cat. #10849) in ddH₂O was evenly distributed across its surface. The dish was incubated for at least 30 min at 50 °C until a dry gel formed. Then, 162 μl of the lipid mixture (typically DOPC with additional 0.5–1% fluorescently labelled DOPE (Atto643 or Atto488) if visualisation by fluorescence was desired, or 0.4% TopFluor cholesterol and LissRhod-PE each for visualisation of lipid phases. Total lipid concentration: 2.54 mM) in chloroform was evenly spread out across the surface using a Drigalski spatula until the chloroform evaporated. Remaining traces of solvent were removed by exposing the dried lipid cake to a vacuum for at least 15 min. Next, 4–5 mL of sodium buffer was added to the petri dish and the solution was left in a humid chamber in the dark at RT for at least 1 h. GVs were detached by gently tapping against the dish and pipetting up and down with a cut pipette tip. The GV suspension was stored at 4 °C in the dark.

When GVs were to be washed and concentrated, the dry lipid film was swollen in caesium buffer or sucrose buffer and, after harvesting, transferred into a Falcon tube. The tube was filled up with sodium buffer (isosmotic to the swelling buffers) and centrifuged for at least 30 min at 300 × g. The supernatant was discarded, and the GV pellet was washed 2–3 more times by adding 900 μl sodium buffer and centrifuging the suspension for 7 min at 300 × g.

Smaller-sized vesicles were prepared using a mini-extruder (Avanti Research) and polycarbonate membranes with the desired pore size (400, 200, 50 nm; Whatman). GV preparations were pushed back and forth through the membrane 21 times.

Lipids in vesicle preparations were quantified using a colourimetric phospholipid quantification assay kit (Sigma Aldrich CS0001) following the manufacturer's instructions. The absorbance was measured at 570 nm on a Clariostar plus microplate reader (BMG Labtech).

Vesicles used in the lipid mixture screens (Fig. 3e, f) were swollen in sucrose buffer at 60 °C to account for differences in lipid charge and melting temperature and washed as described above. Lipid quantity was estimated by measuring the lipid content in sample A (DOPC vesicles) using the colourimetric assay mentioned above. As this assay can only detect choline-containing lipids, the lipid quantity in the remaining samples was estimated based on the fluorescence of Atto643-DOPE species included at equal molar percentages in all lipid mixtures. For this, vesicle preparations were diluted 1:10 and 10 μl aliquots were mixed with 90 μl Triton-X100 (2%) solution in a black 96 well plate. The plate was incubated at 60 °C for 1 h to disintegrate the vesicles, and the fluorescence of Atto643-DOPE was measured in a plate reader. The lipid quantity in the other vesicle preparations was finally calculated by normalising the fluorescence readouts with respect to that of sample A, and multiplying the obtained ratio by the

colourimetric quantification result for sample A. Images of phase-separated vesicles were obtained using a Thermo Fisher EVOS M7000 imaging system.

## Vesicle binding studies

Microscope slides and coverslips were washed with ddH$_2$O and incubated in denatured BSA blocking buffer (10 mM TRIS, 150 mM NaCl, 50 μM BSA, pH 8, heated to approx. 70–90 °C for 30 min until cloudy) overnight at RT on a rocking shaker, as described previously[40]. Next, the glassware was thoroughly washed with ddH$_2$O to remove residual blocking buffer and dried at 50 °C for 1 h. 6 μl of GVs (99.95% DOPC, 0.05% DOPE-Atto488; swollen in imaging buffer A) was mixed with origami triangles (±chol-oligos; fluorescently labelled with Atto643 modified oligonucleotides; final origami concentration: 7 nM) and topped up to 40 μl with imaging buffer B. 300 mM NaCl was added to the samples, testing the effect of elevated sodium on the binding behaviour of origami. The samples were pipetted into an imaging chamber formed by sandwiching a silicon isolator (Grace Bio-Labs) between passivated coverslips and microscope slides. The samples were imaged using a Leica TCS SP5 confocal microscope with a Leica HCX PL APO CS 63×/1.40–0.60 oil immersion objective. Intensity profiles were obtained using Fiji v2.16.0/1.54p[41].

The binding behaviour of origami triangles onto phase-separated vesicles (as shown in Supplementary Fig. 22) was analysed by mixing triangles (15 nM in sodium buffer, chol-hybridised) and GVs (DOPC/DPPC/chol/TF-chol/LissRhod-PE (40.2:40:19:0.4:0.4 mol%)) swollen at 65 °C in sucrose buffer for 1:45 h, followed by 3 washing steps and resuspension into approx. 600 μl sodium buffer) at volume ratios of 1:10 or 1:2 (origami:GVs), and incubating the mixtures for approx. 1 h at RT. 20 μl of the mixtures were then diluted 1:2, adjusting the salt conditions as desired (5 mM or 65 mM MgCl$_2$, 300 mM NaCl). The mixtures were then diluted with approximately. 300 μl buffer of the same composition to reduce background signal from unbound origami, and pipetted into a chamber formed by sandwiching a silicon isolator in between BSA-blocked coverslips and microscope slides. The samples were imaged at RT using a Thermo Fisher EVOS M7000 imaging system with software v2.4.1468.172.

## General budding assay

Giant vesicles were prepared as described above using 99.0–100% DOPC (unless stated otherwise); for fluorescence imaging, a small fraction of fluorescently labelled DOPE was included. Origami triangles were hybridised with chol-oligos at a 1.5x excess relative to the total number of compatible linker handles. The hybridisation reaction was incubated at RT for at least 30 min.

For outward budding, origami triangles bearing linker handles on the shell-inward face were added to 24 μl of untreated GV suspension to a final concentration of ~2 nM. Where lipid quantities were known, GVs corresponding to 1000 pmol (prepared in caesium buffer) or 350 pmol (prepared in sucrose buffer) lipids were combined with 4.5 μl origami at 15 nM. Samples were brought to 30 μl with sodium buffer, and the MgCl$_2$ was adjusted to either 65 mM (by adding 1 M MgCl$_2$ solution (hypertonic)), or to 60 mM (with isotonic assembly buffer). Reactions were incubated at 37 or 40 °C for ≥16 h, up to 3 d, unless stated otherwise.

## Modified budding assays and DCV characterisation

Stability assays under low-salt conditions were performed by dialysing completed budding reactions and vesicle-free controls overnight against sodium buffer (Slide-a-lyzer mini dialysis cups, 20 kDa MWCO, Thermo Fisher). Control samples were taken immediately before dialysis and stored at room temperature.

Quantification of DCV yield as a function of lipid quantity and parent vesicle size was performed using assays with a constant amount of DNA triangles (4.5 μl at 15 nM) and varying lipid quantities (in the form of SVs, LVs, or GVs), as indicated. DCV lipid intensity was measured in Fiji by placing a rectangular selection at the height of the DNA shells, without distinguishing between subtypes (e.g. octahedra vs. icosahedra). The total measured intensity, representing the combined signal from all DCV subtypes, was normalised to the highest intensity observed within each gel, normalised once more to set the largest mean value in each set of triplicates to 1.0, and plotted using GraphPad Prism v10.1.1. Each gel included all tested lipid quantities for a given vesicle type; vesicle types were analysed on separate gels. Experiments were carried out in triplicate on separate days, each using a freshly prepared vesicle batch.

For the tonicity screen, undiluted, chol-hybridised triangles were mixed with GVs and diluted with water and glycine buffer. After a 2 h incubation at RT to allow membrane binding and osmotic equilibration, MgCl$_2$ was added to initiate assembly, and samples were incubated overnight at 37 °C. Detailed sample compositions are provided in Supplementary Table 25.

To study the effect of hypertonic conditions on monomeric triangles, MgCl$_2$ was kept at 5 mM while glycine buffer was added to a total of 700 mM, lowering NaCl from 300 mM to 31 mM. Isotonic controls were kept at 5 mM MgCl$_2$ and 300 mM NaCl without glycine.

For budding via biotin–NeutrAvidin interactions, triangles were folded with linker handles complementary to biotinylated oligos and purification linkers (refer to Supplementary Tables 3, 4 and 20 for sequences). Biotinylated oligonucleotides were included directly in the folding reaction to allow removal of excess strands by gel extraction. The purified triangles were incubated with a 500-fold excess of NeutrAvidin for 1 h at 37 °C on an orbital shaker.

To isolate NeutrAvidin-loaded triangles, the sample was transferred into a BSA-passivated tube and incubated with magnetic beads (Dynabeads M-270 Streptavidin, Invitrogen) coated with dual-biotinylated DNA oligos complementary to the purification handles. Following a 1 h RT incubation on a rotary shaker, the beads (now carrying the origami) were washed 3–4 times with sodium buffer. The triangles were then released from the beads via toehold-mediated strand displacement by adding an excess of invader strand and incubating for 1 h at 37 °C. Then, the NeutrAvidin-bearing triangles were mixed with GVs (96% DOPC, 3% DOPE-biotin, 1% DOPE-Atto488; swollen in caesium buffer, washed and resuspended into sodium buffer), the MgCl$_2$ concentration was adjusted to 60 mM by adding isotonic assembly buffer, and the sample was incubated at 37 °C for 2.5 d.

Budding kinetics were obtained by mixing samples with gel loading dye (50% Ficoll400, 20 mM MgCl$_2$, Orange G) and freezing them in liquid nitrogen at the indicated time points. Once all time points were taken, the samples were thawed, mixed and analysed by agarose gel electrophoresis.

## Inward budding and bivesicular shell assembly

For inward budding, origami triangles (3 μl at 20 nM) with shell-outward linker handles were mixed with LVs (1500 pmol lipids; approx. 200 nm diameter) and incubated at 37 °C for 1–3 d.

Bivesicular DNA shells (VCDCVs) were assembled by successive inward and outward budding reactions using triangles with orthogonal linker handles on both faces. First, DCVs were produced in an outward budding assay (1200 μl in total) using chol-oligos hybridised to the bottom-face linkers. The sample was centrifuged at 300 x g for 5–8 min to pellet residual GVs, and the supernatant was then ultracentrifuged at 55 krpm (avg. 108,000 × g) for 30 min in a Beckman Coulter Optima MAX-TL ultracentrifuge equipped with a TLA-110 fixed angle rotor. 80–90% of the supernatant was removed, and the pellet was resuspended in the remaining volume.

Meanwhile, LVs (~200 nm) were mixed with a second chol-oligo complementary to the top-face linker handles at 50x excess

for 15–30 min at RT. Next, the MgCl$_2$ concentration was adjusted to 60 mM MgCl$_2$ using isotonic assembly buffer. Finally, these LVs (1500 pmol lipids) were mixed with the DCV concentrate (final triangle concentration: 1–5 nM) and incubated at 37 °C for 1–3 d.

### Negative stain transmission electron microscopy

Sample solutions were incubated on glow-discharged copper grids (400 mesh) with a carbon support film (Science Services) for 4–8 min, depending on the sample concentration. The solution was then blotted off using filter paper (Whatman). Then, the grid was washed once using stain solution (2% aqueous uranyl formate +25 mM NaOH), followed by incubating another stain droplet on the grid for 30 s. Excess stain was blotted off using filter paper, and the grid was air-dried before imaging.

Grids were typically imaged at ×26–30k magnification using SerialEM v3.5.6 with an FEI Tecnai T12 electron microscope operated at 120 kV and a Tietz TEMCAM-F416 camera. Image contrast was auto-levelled using Fiji.

For tomography, one-directional tilt series were acquired from −50° to +50° in steps of 2° using SerialEM. The resulting image stacks were imported into Etomo v4.11.24 (IMOD[42]), and images were aligned without fiducials by cross-correlation with cumulative correlation switched on. Tomograms were generated by filtered back-projection (Gaussian filter cut-off: 0.35; fall-off: 0.035). For visualisation, the z-stacks thus generated were imported into Fiji and z-projected by summing up z-slices corresponding to the top, middle, and bottom sections of the respective particle of interest.

### Cryogenic electron microscopy

Sample grids for cryoEM were prepared using an automated vitrification system (FEI Vitrobot Mark V). First, grids (Quantifoil R1.2/1.3, Cu 200 mesh, 100 holey carbon films, 2 nm carbon support film; for images of DCVs: Ted Pella, Cu 400 mesh, lacey carbon, <3 nm carbon support film) were glow-discharged and then inserted into the Vitrobot. The conditions inside the device's environmental chamber were kept at either 4 °C or 22 °C and 100% relative humidity. Samples were typically incubated on the grids inside the chamber for 5 min and plunge-frozen in liquid ethane (blot force: −1, 0 or 25; blot time: 2–3 s; blot total: 1; drain time: 0 s;). The grids were imaged with a spherical-aberration (Cs)-corrected Titan Krios G2 electron microscope (Thermo Fisher) operated at 300 kV and equipped with a Falcon III 4k direct electron detector (Thermo Fisher). EPU v1.2 up to v2.6 (Thermo Fisher) was used for manual data acquisition. Samples were imaged at ×29k magnification and defocus values between −1 and −4 μm. Images were auto-levelled in Fiji.

CryoEM micrographs of membrane buds on LVs were analysed using Kappa v.2.0.0 (Fiji), tracing membrane bud sections underneath triangles and curve fitting to obtain curvature values. Fitting errors were corrected manually.

Cryogenic electron tomography (cryo-ET) samples were mixed with gold nanoparticles (10 nm, OD14, CMC Utrecht), adding 0.5 μl nanoparticle solution per 20 μl sample, directly prior to vitrification. Cryo-ET was performed using the Tomography 5 software suite (Thermo Fisher) on a Titan Krios G2 (300 kV) at low-dose conditions to record a one-directional tilt series (±45° range, steps of 2.5°). Image stacks were processed in Etomo as described for a negatively stained sample, with the difference of using fiducial-aided image alignment. Tomogram contrast was enhanced by Gaussian blurring and auto-levelling images in Fiji.

### Agarose gel electrophoresis and image processing

Agarose gels were cast by dissolving UltraPure agarose (Invitrogen) in running buffer (0.5× TBE, Carl Roth) supplemented with MgCl$_2$ and optionally ethidium bromide (0.025%, Carl Roth).

Samples were run for 2 h at 90 V and then scanned using a Typhoon FLA 9500 laser scanner (GE Healthcare) at a pixel size of 50 μm. Gels containing only monomeric origami structures were run with 5.5 mM MgCl$_2$ in the gel and running buffer and an agarose concentration of 1.5%. To resolve and preserve assembled shell structures, the gel and running buffer were supplemented with 20 mM MgCl$_2$ and the agarose concentration was reduced to 0.8%. Due to the elevated salt concentration in these gels, the gel boxes were cooled in a water-ice bath, and the running buffer was exchanged every 45 min to replenish lost magnesium. For visualisation, brightness/contrast of the area underneath the gel pockets was auto-levelled in Adobe Photoshop and Fiji, intentionally oversaturating the gel pockets in the process.

Gels containing both fluorescent DNA and lipids were scanned using appropriate excitation wavelengths and emission filters. A merged-colour image was created in Fiji, setting DNA signal grey and lipid signal magenta.

### Statistics and reproducibility

The findings in this study have been reproduced $n \geq 2$ times with the same or similar results. All microscopy images presented in the manuscript and the Supplementary Information are exemplary micrographs of larger image stacks representative of the respective sample.

### Reporting summary

Further information on research design is available in the Nature Portfolio Reporting Summary linked to this article.

## Data availability

The data generated in this study are provided within the paper, Supplementary Information files and available from the corresponding author upon request. Unedited gel scans are provided as sourceData1 (PDF). Unedited gel scans, plots of band intensities and raw intensity data underlying Supplementary Fig. 14 are provided as sourceData2 (excel). Source data are provided with this paper.

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

## Acknowledgements

This manuscript has been edited in part with the assistance of the large language models GPT3.5 (openAI) and Perplexity AI. We thank Andreas Bausch and Friedrich Simmel for technical assistance, Björn Högberg for discussions, and Volodymyr Mykhailiuk for help with illustrations. We acknowledge financial support received from the European Research Council Advanced Grant awarded to H.D. (grant agreement 101018465), the Deutsche Forschungsgemeinschaft for the Gottfried Wilhelm Leibniz Program grants provided to H.D., and the Federal Ministry of Education and Research (BMBF) and Bavarian Ministry of Science and Arts through the ONE MUNICH project Munich Multiscale Biofabrication. This project has received funding from the European Union's Horizon 2020 research and innovation programme under the Marie Skłodowska-Curie grant agreement No 765703.

## Author contributions

The research was performed by M.T.P., supervised by H.D. The manuscript was written by M.T.P. and H.D.

## Funding

## Competing interests

The authors declare no competing interests.
