## [Transparent Peer Review file · Nature Communications]

Programmable DNA shell scaffolds for directional membrane budding

Corresponding Author: Professor Hendrik Dietz

Version 0:

Reviewer comments:

Reviewer #1

(Remarks to the Author)

In this manuscript, Pinner and Dietz present a method for creating lipid vesicles of defined sizes via membrane budding mediated by the assembly of DNA origami into octahedra and icosahedra shells - a process analogous to clathrin-mediated budding. They characterize these DNA origami-coated vesicles using gel electrophoresis, fluorescence microscopy and electron microscopy. The authors also demonstrate the fabrication of vesicle-coated DNA origami shells and a nested bivesicular system by manipulating the direction of curvature and positioning of cholesterol moieties. This, in turn, opens up the possibility to create nested vesicles with distinct buffering environments in the future.

While the concept is exciting, the manuscript has several areas that could be improved. In particular, the experimental design is occasionally inconsistent, data analysis of results lacks depth, and additional experiments are required to substantiate some claims.

I believe that implementing the following suggestions will improve the manuscript's quality and its suitability for publication in Nature Communications.

Major suggestions:

1. Could the authors elaborate on how their work differs from the recent work done by Liu and co-workers (2024)? Though the authors reserve priority through their Biorxiv paper, now that the other work has been published, I think it is important for the readers to understand the contrasts and appreciate the highlights of this manuscript. Could the authors include a brief discussion on this?
2. As a follow-up, it would benefit the readers if the authors could provide more context and discussion of their design choices in the main text rather than placing this information in the legends in Supplementary figures.
3. The authors note "The representative tomograms confirm full engulfment of the vesicle by closed shells featuring the symmetry properties expected by design." However, the representative z-slices from the tomogram appear less convincing possibly due to flattening of samples and imaging artefacts in ns-EM (as noted by the authors in Supplementary Figure 9). I encourage the authors to perform cryo-ET on the samples and to include the reconstructed tomogram(s) as a supplementary to better validate the results.
4. The authors state "cholesterol insertion is not required for DCV formation, supporting the idea that membrane buds are primarily shaped by shell assembly." While it is not required for DCV formation, the data suggests that chol-oligo-containing origamis drive the population toward octahedral structures, whereas NAv appears to retain the icosahedra shell population. Could the authors comment on this in the main text?
5. The authors employ both Atto488- and Atto643-modified DOPE. Is there a specific rationale for this? It creates confusion and the work would be consistent if the authors used only Atto488, since DNA origami already is labelled with Atto643. For example, in Supplementary Figure 13, the authors state they imaged in the lipid channel using Atto643-DOPE. Considering that the origamis also have Atto643, it might be helpful to use Atto488 -DOPE (as in a few other supplementary figures) to demonstrate the GV co-localization signal and further demonstrate the difference in origami signal from co-localized GVs

between M5 and M60 variants. Moreover, in the 'Vesicle preparation and quantification' in the methods, the authors only describe Atto643-DOPE preparation. Could authors clarify whether Atto488-DOPE was also prepared the same way in all the experiments performed with Atto488-DOPE?

6. In Supplementary Figure 3, the authors state "Reducing the reach of individual cholesterols by positioning them closer to the centre of the triangles and choosing proximal configurations reduces unwanted interactions." Which variant does this statement correspond to? Does it correspond to the Variant Bp? Please include the results for the variant Bp. Additionally, please include a discussion of different variants in the main text, as the difference between proximal and distal binding is not often discussed and could provide interesting insights to the reader.

7. As a follow-up, for Supplementary Figure 4 and other subsequent figures, please indicate which variant was used AD, AP, BD or BP? Additionally, how do the authors confirm that specific gel bands correspond to octahedra or icosahedra structures? Was this verified by excising the bands and imaging them? Furthermore, could the authors clarify the band in between icosahedra and octahedra in the 1 chol-oligo/triangle variant?

8. There seem to be inconsistencies between Supplementary Figure 4 (1, 3 and 9) and 5 (3, 6 and 9) in the number of chol-oligo/triangles. Could the authors elaborate on what happens in the presence of only one chol-oligo/triangle in the presence of GVs? Additionally, was the gel visualized using the Atto643 signal from origami, and were the GVs labelled with any fluorophores?

9. In Supplementary Figure 7, could the authors clarify the reason for smaller vesicles in the presence of NaCl without chol-oligo? Is the presented image a representation? Could the authors provide statistics on vesicle sizes for different conditions? Do they change and is there a correlation? Additionally, in Supplementary Figure 8, the authors state "DCVs make up the largest particle fraction". Could the authors include statistics of different particle fractions to substantiate this statement? Please include more quantitative comparisons and statistical analyses throughout the manuscript to help the readers understand the data better.

10. In Supplementary Figure 12, could the authors show what happens when the Mg concentration is increased beyond 65 mM? Considering icosahedra species start to appear at 65 mM Mg in the non-passivating version, could the authors compare higher Mg concentrations (>65 mM Mg) using a gel? Furthermore, please comment on why the shell assembly in this work deviates from the typical threshold concentration for shell assembly (also, please cite the reference that mentions the typical concentrations for shell assembly).

11. As a follow-up, the band for monomeric triangles seems to be depleted at 65 mM Mg in the non-passivating version. Does increasing the concentration of monomeric triangles enhance DCV formation? If it does, which population will be enhanced, octahedra or icosahedra? Could the authors please show some data with increased concentration of the monomeric triangles?

12. The gel image in Supplementary Figure 13 appears to be cropped as the monomeric and oligomeric triangle bands are not visible. Could the authors include the full gel image? Additionally, the authors state "We finally obtained stronger lipid signals in samples initially incubated at 5 mM", is there a reason for using the word 'finally' here? Could the authors show what happens when the concentration of Mg is lowered after the formation of DCVs? Does the shell fall apart leaving the vesicles or does the buried cholesterol keep the origami shell integrated?

Minor suggestions:

1. To help grow the origami community and contribute to open research, I suggest the authors submit their origami designs to nanobase.org.

2. In Fig 2d, the authors talk about the influence of orthogonal chol-oligos. Please provide an illustration indicating the location of these orthogonal chol-oligos (similar to Fig. 4a). Are these origamis with chol-oligos? Or are they just excess origami-free chol-oligos which are used in the bivesicular system? Could the authors clarify this?

3. As a follow-up, in Supplementary Figure 4, which positions were chosen for placement of 1 and 3 chol-oligos (only the 9 chol-oligo illustration was shown in Supplementary Figure 3)? Could the authors provide some illustrations for the position of chol-oligos in Supplementary Figure 4?

4. The authors state that "spontaneous membrane scission outcompetes the completion of the DNA origami shell leading to the formation of DCVs" which in some cases leaves a residual "scar". Could the authors comment on what prevents the excess monomeric triangles from completing the shell after budding? Could this relate to the reduced intensity of monomeric and oligomeric origami bands at 65 mM Mg for the non-passivating variant (refer to major suggestions)? How many instances of "scars" did the authors observe in their data? Please include some statistics.

5. The term "vesicle-coated DNA shells" seems somewhat confusing. I think DNA origami-coated vesicle is apt since individual origami structures assemble into a coating/shell. However, the vesicles do not coat/assemble on the surface of origami but rather the origami shell formation leads to their engulfment/invagination/encapsulation by vesicles. Could the authors comment on this and make the naming more intuitive?

6. The manuscript mentions "... solution at 37 °C for up to multiple days". Could the authors specify the duration more specifically or provide a range of days as they do in the methods section?
7. Please indicate which channels were used for imaging for all gels (missing in some gel images like Supplementary Figures 3 and 4).
8. Please denote 'BD' in the figure legend for the distal positioning of cholesterol closer to the centre of the structure in Supplementary Figure 3. Additionally, please indicate how many chol-oligos were used in Supplementary Figure 3c. Finally, please use the word 'chol-oligos' instead of 'chol' in the gel images as that has been the terminology used throughout the manuscript (in Supplementary Figure 3 and for other figures as well).
9. Although the reader can interpret upon careful reading, could the authors please clarify that the origami signal in the gel pocket of Supplementary Figure 11 (and some other images as well) might represent the GV bound triangles in monomeric/oligomeric form which were unable to form the DCVs and so doesn't enter the pocket? Additionally, please clarify that the monomeric and oligomeric origami signal in the Atto643 channel could correspond to the excess origami which might not be bound to GVs.
10. In Supplementary Figures 16 and 18, it appears as though the gels were run at higher magnesium concentration, as one could see a band corresponding to octahedral in the monomer lane. Please clarify this and ensure that all figure legends should include the gel conditions wherever they deviate from standard running conditions.

Reviewer #2

(Remarks to the Author)

This manuscript reports the generation of budding vesicles with self-assembling DNA triangles modified with membrane anchors. Biomimetic DNA origami structures, mostly built to model the activities of BAR domains, dynamin, ESCRT and clathrin, have been reported to deform lipid membranes. Compared with previous studies, this work stands out because it (1) achieved well-defined self-assembling-dependent vesicle budding, (2) thoroughly investigated the design and experimental parameters that influences the deformation outcome, and (3) generated topologically controlled inward and outward-budding events. The membrane budding products and intermediates were visualized by negative-stain EM and cryoEM, the yield of the budding vesicles were estimated by gel electrophoresis. The high-quality data clearly support curvature induction by the self-assembling DNA triangles. The manuscript is well written, and I read it with great interest. Overall, this work highlights the programmability of the DNA-origami-based membrane-deforming platform and is a valuable addition to the high-precision membrane engineering toolbox.

Below are a few questions and suggestions:

- (1) The authors investigated the effect of cholesterol number and positioning on the membrane deformation outcome. However, the copy number and positioning of cholesterol on DNA triangle are not entirely clear for each experiment (e.g., Supplementary Fig 4, 5, 6). Please clarify this for each piece of relevant data presented.
- (2) Budding happens more readily on low-tension membranes, as shown in Fig. 2c. The question is whether the budding under such conditions (i.e. hypertonic) still depends on the self-assembly of DNA shells (i.e., high Mg²⁺ and shape-complementary interfaces)?
- (3) It is remarkable that in certain EM images, the DNA triangles almost exclusively localizes to the membrane buds, suggesting that complete membrane coverage by DNA structures may not be required when DNA triangles assemble into cages. The authors cited ref 14 to argue that attaching DNA origami structures to lipid vesicles can deform vesicles without requiring subunit assembly. However, this study (and many others such as ref 16) show that such deformation requires high surface coverage. It would be nice if the authors can elaborate on this point in the discussion.
- (4) Membrane binding DNA triangles modified by either cholesterol or biotin-neutravidin can similarly generate DNA-coated vesicles. But are they as efficient as each other? How does the yield compare to one another when all else (osmolarity, surface coverage, membrane-anchor positioning, etc.) are equal?
- (5) Cholesterol-labeled DNA structures have been shown to partition in different domains of phase-separated lipid bilayers under certain conditions (e.g., Kanwa 2023 10.1002/admi.202202500). Did the authors observe such phenomena? Did these membrane domains deform similarly or differently?
- (6) In theory, the membrane budding process can be halted or even reversed by lowering the Mg²⁺ concentration, which stops and disrupts DNA shell assembly. Have the authors attempted this? I ask out of curiosity and would not regard the work as incomplete if this piece is missing.

Reviewer #3

(Remarks to the Author)

Pinner et al. presented a DNA origami assembly system on lipid vesicles to mimic the virus assembly and control directional

membrane budding. By leveraging the programmability of DNA origami triangles, this approach enables the formation of DNA-shell-coated vesicles, vesicle-coated DNA shells, and multivesicular DNA shells. The resulting DNA origami-based membrane budding system in some manner replicates key aspects of natural endocytic and exocytic pathways. Overall, the study presents clear and compelling results, making it highly relevant to researchers in DNA nanotechnology and synthetic biology. Especially, the presented structures are of exceptional quality, as always from Dietz's lab.

- While the authors have previously reported the assembly of DNA origami triangles into icosahedral shells (Reference 21), and the impact of cholesterol decoration on individual origami triangles and their assembly has been demonstrated via agarose gel electrophoresis (Figure S3), this evidence alone is insufficient to substantiate the claim in Line 78 that "and then validated their assembly into the expected icosahedral shells in the absence of membranes". Additional TEM images would help validate this statement. Furthermore, for the discussions related to Figures S4-S6, additional TEM characterization is recommended to complement the agarose gel data and strengthen the analysis.

- For the key TEM images that support the main results of this study (Figure 1d, Figure 3b and Figure 4b), only some cropped TEM images are presented. It is necessary to include corresponding overview TEM images to estimate the yield of this artificial membrane budding system.

- Regarding the schematic in Figure 1a, the membrane appears to be depicted as a 2D surface. However, all membrane budding experiments described in the manuscript were conducted on 3D lipid vesicle membranes. Therefore, it is recommended to revise Figure 1a to represent a 3D membrane instead of a 2D one.

- In the first section of "DNA-shell-coated vesicles form by membrane budding," the conclusion in Line 381 that "curvature increases gradually and closely follows the intrinsic curvature of the assembling shells" and the hypothesis in Line 138 that "Considering that the increasingly high curvature at the bud neck can become a steric barrier to completion of the DNA origami shell, we believe the occurrence of scars to be a signature of the proposed budding process" are drawn primarily from the TEM images in Figures 1d and 2b. However, these images alone provide limited evidence to fully support these claims. To substantiate the statement that "curvature increases gradually and closely" and the hypothesis regarding "the increasingly high curvature at the bud neck," real-time monitoring of the 2D membrane using AFM would be a valuable approach. Alternatively, a statistical analysis of curvature based on similar TEM features observed at different stages, as shown in Figure 3c, could also strengthen the argument.

- Is it possible that the DNA origami triangles first assemble in bulk before being loaded onto vesicles? Long-term AFM monitoring on a 2D membrane may also provide insights into this process.

- In Figure 1b and Figure S7, the authors report that "At low-salt conditions, triangles adsorb onto lipid vesicles even if not hybridised to chol-oligos. By adding 300 mM NaCl, non-specific origami-vesicle association is suppressed without interfering with cholesterol-mediated association." However, the exact NaCl concentration in these "low-salt conditions" is unclear. Based on the figure caption of Figure S7, it appears to be zero. Additionally, the rationale behind selecting a NaCl concentration of 26.3 mM in Supplementary Table 25 and 5 mM in the imaging buffer and isotonic assembly buffer in Supplementary Table 26 requires clarification. Given that these values are significantly lower than 300 mM, it is important to explain how non-specific binding is mitigated under such low-salt conditions. These aspects should be addressed in both the manuscript and the Supplementary Information.

- The effect of DNA origami triangle density or quantity on individual vesicles in relation to membrane budding should be experimentally verified.

- The authors summarize in Line 284: "In summary, we present a versatile platform for engineering vesicles of controlled size featuring addressable endo- or exoskeletons." However, given that the lipid vesicle sizes shown in Figure 1b vary over a broad range, it is also recommended to investigate and demonstrate the impact of vesicle size on membrane budding.

Minor issues:

- In line 34, the results on DNA origami in this study do not fully support the statement: "We hypothesized that any material capable of self-assembly into curved shapes could act as a molecular scaffold for membrane budding," It is recommended to clarify this statement to better reflect the findings.

- Typically, large-sized lipid vesicles are referred to as "giant unilamellar vesicles (GUVs)," as mentioned in Line 45. To maintain consistency, it is suggested to replace "giant vesicles (GVs)" and "large vesicles (LVs)" with "giant unilamellar vesicles (GUVs)" and "large unilamellar vesicles (LUVs)" throughout the manuscript.

- There are two instances of the abbreviation "(DCV)" in Lines 61 and 74. Additionally, the abbreviation "(VCD)" appears in Lines 61, 74, 219, and 418.

- The experimental description should be more detailed. For example, in Line 99, the statement: "We then triggered the assembly of membrane-bound triangles by increasing the MgCl₂ concentration and incubating the solution at 37 °C for up to multiple days" lacks specificity. It would be clearer to specify the final Mg²⁺ concentration and replace "multiple days" with a precise duration (e.g., 48 or 72 hours).

- The manuscript includes only two references published after 2023. It is suggested to update the references and incorporate discussions on more recent advancements in DNA nanotechnology and synthetic biology to ensure the work is aligned with

the latest research.

Version 1:

Reviewer comments:

Reviewer #1

(Remarks to the Author)

The manuscript has improved substantially post-revision. The authors have performed several additional experiments to validate their results and substantiate their claims as reflected in the changes in main figure 2 and supplementary figures 3, 4, 11, 12, 13, 14, 16 and 17. The authors have clarified my questions and implemented the recommended suggestions.

I recommend publication after addressing some minor comments.

1. Please include a scale bar in Supplementary Figure 17. Furthermore, please include the scale bar in the caption in Supplementary Figures 3 and 23. Also, please include the caption for Supplementary Figure 3d.
2. Are there additional representative images for Supplementary Figure 7?
3. Would the authors consider sorting buds 1-27 as increasing curvature from top to bottom in Supplementary Figure 10? In this way, the process of budding can be seen as a gradual increase in bud curvature and would be more intuitive.
4. Just out of curiosity, is there a reason for redundant figures in the main (2a and 3a) and supplementary (SF 11d and 17a)?

Reviewer #2

(Remarks to the Author)

The authors have adequately addressed all my points. I support publication enthusiastically.

Reviewer #3

(Remarks to the Author)

The authors have sufficiently addressed my comments.

Reply to reviewers

We thank the reviewers for their fair and constructive comments on our work. Addressing their questions has helped us improve the manuscript and substantiate conclusions drawn. Below we provide a point-by-point response to each comment raised.

Reviewer 1

In this manuscript, Pinner and Dietz present a method for creating lipid vesicles of defined sizes via membrane budding mediated by the assembly of DNA origami into octahedra and icosahedra shells - a process analogous to clathrin-mediated budding. They characterize these DNA origami-coated vesicles using gel electrophoresis, fluorescence microscopy and electron microscopy. The authors also demonstrate the fabrication of vesicle-coated DNA origami shells and a nested bivesicular system by manipulating the direction of curvature and positioning of cholesterol moieties. This, in turn, opens up the possibility to create nested vesicles with distinct buffering environments in the future.

While the concept is exciting, the manuscript has several areas that could be improved. In particular, the experimental design is occasionally inconsistent, data analysis of results lacks depth, and additional experiments are required to substantiate some claims.

Thank you very much for your thoughtful and detailed review of our manuscript. We greatly appreciate your insightful comments that have helped us clarify and strengthen our work. In particular, we are grateful for your encouragement to expand on our design choices and improve statistical analysis, which resulted in a new main figure. We have also enhanced our description of the morphology of our particles with cryoET reconstructions to complement our ns-ET data, as per your suggestion. We believe that your constructive feedback has been invaluable in improving the clarity and rigour of our manuscript.

Comment 1-1

Could the authors elaborate on how their work differs from the recent work done by Liu and co-workers (2024)? Though the authors reserve priority through their Biorxiv paper, now that the other work has been published, I think it is important for the readers to understand the contrasts and appreciate the highlights of this manuscript. Could the authors include a brief discussion on this?

Prior to publication, we were in contact with Prof. Liu to discuss potential overlaps, and we mutually agreed that the similarities between the studies are limited.

Their work primarily investigates microscopic membrane deformation and pore formation. While they present DNA-lipid hybrid structures resembling our DNA-shell coated vesicles (DCVs) in Fig. 5C/D, no experimental evidence is provided to support a budding mechanism. The term "budding" is mentioned only once in the introduction and is not used to describe their process.

In contrast, our study focuses on the mechanism of DCV/VCD formation. Using cryo-EM, we directly visualise nanoscopic membrane deformation and demonstrate that the formation efficiency is modulated by membrane tension, an essential feature shared with natural budding processes such as clathrin-mediated endocytosis. Our findings support the conclusion that DCVs form via a membrane budding mechanism.

To clarify this distinction, we have added the following lines to the discussion:

Lines 369-373:

(...) In a related study, Zhan, Yang et al. extracted membrane material from donor vesicles using cholesterol-decorated DNA origami structures, but mechanistic details regarding the underlying process remain to be elucidated³³. Our work complements this study by demonstrating the universality of the molecular scaffolding concept, and identifies budding as the underlying process behind DCV formation. (...)

Comment 1-2

As a follow-up, it would benefit the readers if the authors could provide more context and discussion of their design choices in the main text rather than placing this information in the legends in Supplementary figures.

We have expanded the main text to include a more thorough explanation of our design choices concerning the nature, positioning, and distribution of linker handles.

Lines 75-78:

(...) Single-stranded linker-handles protruding from the bottom face of the triangle served as attachment sites for chol-oligos (Supplementary Figure 3a). We distributed them across the surface, keeping their relative position on each triangle side consistent and generally aiming for a symmetric layout. (...)

Lines 88-93:

(...) We identified both the proximity of linker handles to the outer edges and the chol configuration (proximal vs distal orientation) after hybridisation to a linker handle as influencing variables. By limiting the reach of each chol moiety (e.g. by increasing the separation of chol between neighbouring triangles in a shell assembly), we could largely revert the assembly to predominantly icosahedral structures and also reduce chol-mediated aggregation. (...)

Comment 1-3

The authors note "The representative tomograms confirm full engulfment of the vesicle by closed shells featuring the symmetry properties expected by design." However, the representative z-slices from the tomogram appear less convincing possibly due to flattening of samples and imaging artefacts in ns-EM (as noted by the authors in Supplementary Figure 9). I encourage the authors to perform cryo-ET on the samples and to include the reconstructed tomogram(s) as a supplementary to better validate the results.

We have performed cryo-ET on DCVs to further validate our structural interpretation. The resulting data are now included in Supplementary Figure 17. We acquired tilt series using a low electron dose to reduce radiation damage, which necessarily resulted in low contrast in individual images and tomogram slices. To improve visualisation, we summed slices corresponding to the top, middle, and bottom sections of the DCV (consistent with our approach in the ns-ET data) and applied a mild Gaussian blur. In addition, we provide an animated .gif of the tilt series used for reconstruction, which clearly shows the (scarred) DCV from multiple angles (Supplementary Data 1).

The cryo-data confirms vesicle engulfment, and due to the scar in the shell, the vesicle remains partially visible in the bottom region. For comparison, we also include cryo-ET sections of an empty shell.

Supplementary Figure 17 | Tomographic analysis of DCVs and an empty shell. a, Summed z-slices from tomograms of a DCV showing top, central, and bottom sections. Left: negatively stained electron tomography sample (ns-ET); right: ET of a vitrified sample (cryo-ET). Cryo-ET preserves the native structure but offers lower contrast, while ns-ET provides higher contrast at the cost of potential artefacts such as vesicle distortion. A gap in the DNA shell ("scar") is visible in the bottom slice (arrow), revealing the enclosed vesicle. In the top slice, limited contrast prevents resolution of individual triangles. The particle appears slightly tilted relative to the schematic at left, with a triangle atop the structure rather than a pentameric cap. **b**, Cryo-ET of an empty shell. Dark spots are 10 nm gold nanoparticles used for tomogram alignment. Images are Gaussian-blurred to enhance contrast.

Comment 1-4

The authors state "cholesterol insertion is not required for DCV formation, supporting the idea that membrane buds are primarily shaped by shell assembly." While it is not required for DCV formation, the data suggests that chol-oligo-containing origamis drive the population toward octahedral structures, whereas NAv appears to retain the icosahedra shell population. Could the authors comment on this in the main text?

We suspect that electrostatic interactions between the DNA origami and the membrane, as observed in Fig. 1b, may contribute to these geometry-altering effects. To address this, we have added the following explanation to the main text.

Lines 261-265:

(...) Compared to DCVs obtained using chol, the distribution between icosahedral and octahedral DCVs is more balanced when using b-NAv, though overall DCV yields appear to be slightly lower. We hypothesise that a sufficiently high density of NAv on the membrane may act as a protective barrier between both components, increasing their distance from one another and thereby altering electrostatic influences. (...)

Comment 1-5

The authors employ both Atto488- and Atto643-modified DOPE. Is there a specific rationale for this? It creates confusion and the work would be consistent if the authors used only Atto488, since DNA origami already is labelled with Atto643. For example, in Supplementary Figure 13, the authors state they imaged in the lipid channel using Atto643-DOPE. Considering that the origamis also have Atto643, it might be helpful to use Atto488-DOPE (as in a few other supplementary figures) to demonstrate the GV co-localization signal and further demonstrate the difference in origami signal from co-localized GVs between M5 and M60 variants. Moreover, in the 'Vesicle preparation and quantification' in the methods, the authors only describe Atto643-DOPE preparation. Could authors clarify whether Atto488-DOPE was also prepared the same way in all the experiments performed with Atto488-DOPE?

We generally prefer Atto643 due to its excellent water solubility, brightness, and low background. It was initially used to label DNA origami because its negative charge and hydrophilicity reduce the risk of membrane insertion. When we needed to label both vesicles and origami, we used Atto488-DOPE to avoid spectral overlap.

Our use of both dyes reflects the historical development of the project. We later adopted Atto643-DOPE for certain experiments—such as the lipid mixture screen—because it performs better in gel-based detection, where green autofluorescence from agarose interferes with Atto488 signals. In these cases, DNA was visualised with ethidium bromide.

In Supplementary Figure 16 (formerly 13), we show only the lipid channel (Atto643-DOPE) to highlight differences in lipid signal intensity. At no point did we label both the DNA and lipids with the same dye or overlapping fluorophores.

We have clarified this in the Methods by explicitly noting that both Atto643 and Atto488-DOPE were used, and that vesicle preparation was identical regardless of fluorophore:

Lines 700-703:

(...) Then, 162 µl of the lipid mixture (typically DOPC with additional 0.5-1% fluorescently labelled DOPE (Atto643 or Atto488) if visualisation by fluorescence was desired, or 0.4% TopFluor cholesterol and LissRhod-PE each for visualisation of lipid phases. (...)

In the final paragraph of the section, we refer specifically to Atto643-DOPE as it pertains to a particular experiment (the lipid mixture screen) in which Atto488-DOPE was not used.

We acknowledge that using a consistent labelling scheme throughout the manuscript would have been clearer for the reader, and we apologise for any confusion caused. Nonetheless, we hope this explanation clarifies the rationale behind the choices made and how they served the differing experimental needs.

Comment 1-6

In Supplementary Figure 3, the authors state "Reducing the reach of individual cholesterol molecules by positioning them closer to the centre of the triangles and choosing proximal configurations reduces unwanted interactions." Which variant does this statement correspond to? Does it correspond to the Variant Bp? Please include the results for the variant Bp. Additionally, please include a discussion of different variants in the main text, as the difference between proximal and distal binding is not often discussed and could provide interesting insights to the reader.

The statement refers to our general finding that both handle positioning and cholesterol configuration affect assembly. Positioning handles near the triangle centre (variant B) and using a proximal configuration (A_P, B_P) reduce unwanted interactions.

We have now included results for variant B_P and expanded the main text discussion to highlight the influence of these design parameters.

Lines 87-95:

Introducing chol resulted in a shift from the previously observed relatively uniform icosahedral shells to a mixture containing both octahedral and icosahedral shells. We identified both the proximity of linker handles

to the outer edges and the chol configuration (proximal vs distal orientation) after hybridisation to a linker handle as influencing variables. By limiting the reach of each chol moiety (e.g. by increasing the separation of chol between neighbouring triangles in a shell assembly), we could largely revert the assembly to predominantly icosahedral structures and also reduce chol-mediated aggregation. We observed the fewest chol-mediated interactions using triangles with centrally-placed linker handles & proximal chol (Supplementary Figure 3b & c), and we have thus used this setup in all subsequent experiments.

Supplementary Figure 3 | The influence of cholesterol position and orientation on origami triangles. **a**, Linker handle positions on two different triangle variants, and scheme of distal and proximal cholesterol configurations. The upper, white helix represents the triangle surface (side view), the pink strand represents the cholesterol-bearing oligo (chol-oligo). Cholesterol may be oriented facing away (distal) or towards (proximal) the origami surface. **b**, The overall reach of chol-oligos determines the migration behaviour of origami triangles. When cholesterols are positioned closer to the outer edge and the stacking contacts in a distal configuration (A_D), bands in an agarose gel appear shifted and smeared compared to the same triangles hybridised to unmodified (chol-free) oligos. This effect could be reduced by changing the orientation of the cholesterol to a proximal configuration with respect to the origami, such that the reach of the cholesterol is minimised (A_P). Chol-mediated interactions could be reduced further by positioning the linker handles closer to the centre of the structure, and thus further away from the stacking contacts (B_D & B_P). However, chol configuration appears to have a stronger effect than handle placement, and the order of triangle variants from most to least chol-mediated interactions is thus $A_D > B_D > A_P > B_P$. **c**, The reach of cholesterol influences the assembly of cholesterol-decorated triangles. Triangles hybridised to unmodified oligos assemble mostly into icosahedral species, but when chol-oligos are introduced, distal cholesterol (A_D , B_D) promotes the formation of primarily octahedral species and aggregates. Proximal cholesterol (A_P , B_P) does not noticeably influence the assembly behaviour, with most triangles forming icosahedra. As in b, the least chol-mediated interactions were observed in sample B_P , showing that reducing the reach of individual cholesterols by positioning them closer to the centre of the triangles and choosing proximal configurations reduces unwanted interactions.

Comment 1-7

As a follow-up, for Supplementary Figure 4 and other subsequent figures, please indicate which variant was used AD, AP, BD or BP? Additionally, how do the authors confirm that specific gel bands correspond to octahedra or icosahedra structures? Was this verified by excising the bands and imaging them? Furthermore, could the authors clarify the band in between icosahedra and octahedra in the 1 chol-oligo/triangle variant?

We have added schematics indicating the triangle variant used (cholesterol placement) to all relevant supplementary figures. Following the results in Supplementary Figure 3, all subsequent experiments used triangles with centrally placed linker handles and proximal cholesterol (analogous to variant B_P). This is now clarified in the main text.

Lines 93-95:

(...) We observed the fewest chol-mediated interactions using triangles with centrally-placed linker handles & proximal chol (Supplementary Figure 3b & c), and we have thus used this setup in all subsequent experiments.

Band identities were confirmed by gel extraction and TEM. The intermediate band between icosahedra and octahedra corresponds to icosahedral caps (pentamers) and similar early-stage assemblies. This has been added to Fig. 2b.

Fig. 2 (...) **b**, Identification of shell subspecies by gel extraction and TEM analysis. Electrophoretic mobility reflects particle size: icosahedral shells migrate slowest (top band), octahedral fastest (bottom band), with occasional intermediates (e.g. pentamers, half-shells) in between. Monomeric triangles run below assembled shells but are often too faint to detect. DCV formation can be assessed by agarose gel electrophoresis using fluorescently labelled DNA or ethidium bromide (white) and a small fraction of labelled DOPE lipids in the vesicle mixture (magenta). Lipid signal colocalises only with assembled shells, indicating that membrane material migrates through the gel primarily as part of a DCV. DNA retained in the gel pocket reflects vesicle-bound triangles; the presence of a monomer band indicates saturated membranes. Pockets were oversaturated to visualise faint lower bands. Scale bars: 100 nm.

Comment 1-8

There seem to be inconsistencies between Supplementary Figure 4 (1, 3 and 9) and 5 (3, 6 and 9) in the number of chol-oligo/triangles. Could the authors elaborate on what happens in the presence of only one chol-oligo/triangle in the presence of GVs? Additionally, was the gel visualized using the Atto643 signal from origami, and were the GVs labelled with any fluorophores?

We have expanded Supplementary Figure 4 to include TEM micrographs and added the lipid-channel gel results (DOPE-Atto488-labelled GVs) for clarity.

Triangles bearing a single cholesterol can still induce DCV formation, but we frequently observed shells containing vesicles not completely filling up the cavity. For this reason, we chose not to go below three cholesterol in subsequent experiments, including those in Supplementary Figure 5. We instead focused on a symmetric arrangement of cholesterol, distributing 3, 6, or 9 moieties per triangle (i.e. 1, 2, or 3 per triangle side).

Supplementary Figure 4 | The influence of cholesterol count per triangle in assembly reactions containing GVs. **a**, Cholesterol positions on the bottom face of the triangles used in this experiment. **b**, Agarose gel of assembly reactions including triangles carrying either 1, 3 or 9 cholesterols and fluorescently-labelled GVs. Left: Gel scanned for GelRed, visualising DNA. Right: Gel scanned for Atto488, visualising DOPE-Atto488 labelled membranes. Whilst optimising cholesterol positioning on the triangle can reduce unexpected assembly behaviour, introducing lipid vesicles yet again shifts the assembly towards octahedra, the degree of which increases with the number of cholesterols per triangle. The colocalization of lipid and DNA bands suggests the formation of lipid-DNA hybrid structures. **c**, TEM images confirm the formation of lipid-DNA hybrid structures in all samples. However, whilst lipid vesicles commonly fill up the inner cavity of the DNA shells in the 3 & 9 chol / triangle samples, this was not always the case in the 1 chol / triangle sample. Scale bars: 100 nm.

Comment 1-9

In Supplementary Figure 7, could the authors clarify the reason for smaller vesicles in the presence of NaCl without chol-oligo? Is the presented image a representation? Could the authors provide statistics on vesicle sizes for different conditions? Do they change and is there a correlation? Additionally, in Supplementary Figure 8, the authors state "DCVs make up the largest particle fraction". Could the authors include statistics of different particle fractions to substantiate this statement? Please include more quantitative comparisons and statistical analyses throughout the manuscript to help the readers understand the data better.

For Supplementary Figure 7, we used two GV batches prepared with or without NaCl in the swelling buffer. We did not observe consistent morphological differences between conditions; the vesicles shown are therefore representative. MgCl₂, included for origami stability, occasionally led to vesicle aggregation, as seen in the third row of SF7.

While we acknowledge the value of detailed size analysis, characterising GV size distributions under varying conditions falls outside the scope of this study, which focuses on nanoscopic budding. The PVA-swelling method and resulting vesicle characteristics are described in reference 25 and related publications studying giant vesicle formation.

We have added statistics of particle subspecies as a function of chol-count per triangle into main Fig. 2a and Supplementary Figure 11.

Fig. 2 | Analysis of DNA shell and DCV subspecies. **a**, Relative abundance of particle subspecies as a function of cholesterol count. In addition to octahedral and icosahedral DCVs, pentagonal DCVs—likely representing incomplete ("scarred") icosahedra—were frequently observed across all samples. Lower cholesterol content led to more empty shells, while higher cholesterol numbers promoted octahedral DCV formation. Triangles with 3 chol/triangle also yielded a notable fraction of icosahedral shells containing small vesicles ("small-vesicle DCVs"), suggesting weaker membrane binding under these conditions. While these could be staining artefacts, their enrichment at low cholesterol supports a genuine trend. Particles with unclear morphology were excluded. Numbers in brackets indicate absolute particle counts. (...)

Supplementary Figure 11 | DCVs obtained from triangles with 3, 6 or 9 cholesterol. Full fields of view from negatively stained TEM micrographs of DCV samples formed using triangles with **a**, 3; **b**, 6; or **c**, 9 cholesterol. Irregular particles (e.g., unusually large or ambiguous shapes) and free monomers or oligomers were not marked. All images were acquired under identical conditions. Scale bars: 100 nm. **d**, Frequency of particle subspecies by cholesterol count. Besides octahedral and icosahedral DCVs, pentagonal DCVs—likely representing scarred icosahedra—were consistently observed. Lower cholesterol content resulted in more empty shells, while higher counts promoted the formation of octahedral DCVs. Triangles with 3 cholesterol also produced a distinct population of incomplete DCVs with small internal vesicles, less prevalent at higher cholesterol numbers. Although some may be staining artefacts, their enrichment in low-cholesterol samples suggests weaker membrane binding under these conditions. Particles with ambiguous or unclear morphology were excluded from the analysis. Numbers in brackets indicate absolute particle counts.

Comment 1-10

In Supplementary Figure 12, could the authors show what happens when the Mg concentration is increased beyond 65 mM? Considering icosahedra species start to appear at 65 mM Mg in the non-passivating version, could the authors compare higher Mg concentrations (>65 mM Mg) using a gel? Furthermore, please comment on why the shell assembly in this work deviates from the typical threshold concentration for shell assembly (also, please cite the reference that mentions the typical concentrations for shell assembly).

We have added a new Supplementary Figure screening magnesium concentrations between 20-100 mM, and added a few sentences in the main text to explain the results.

Lines 182-188:

As with free triangles, efficient on-membrane assembly requires an optimal magnesium concentration (Supplementary Figure 12). Magnesium modulates electrostatic repulsion and assembly kinetics: low Mg^{2+} impairs shell formation, whereas excess Mg^{2+} stabilises oligomeric intermediates and depletes available monomers. However, membrane-bound triangles appear less sensitive to excess magnesium, likely because their restricted lateral diffusion and the need to induce curvature reduce the formation and release of immature shell intermediates. (...)

Supplementary Figure 12 | Magnesium screen of membrane-bound and vesicle-free assembly reactions. **a**, Agarose gel of triangle assembly on GV membranes. While DCVs may also form at lower magnesium concentrations, assembly is most efficient between 60–70 mM MgCl_2 (in 300 mM NaCl). Monomeric triangles remain bound to GV membranes which are retained in the gel pockets. Faint bands of sandwich dimers—cholesterol-mediated dimers—are visible even at low Mg^{2+} levels. Above the assembly optimum, DCVs form slightly less efficiently. **b**, Agarose gel of vesicle-free triangle assembly. At the magnesium optimum, triangles assemble efficiently into icosahedra, but above and below it, a mix of octahedra and intermediates dominates.

We have also added a sentence to explain the increased magnesium requirements of our system, and added the citation as suggested.

Lines 129-131:

(...) Compared to previous work where triangles were assembled at low NaCl concentrations²², our setup requires significantly higher concentrations of Mg^{2+} to outcompete sodium for assembly. (...)

Comment 1-11

As a follow-up, the band for monomeric triangles seems to be depleted at 65 mM Mg in the non-passivating version. Does increasing the concentration of monomeric triangles enhance DCV formation? If it does, which population will be enhanced, octahedra or icosahedra? Could the authors please show some data with increased concentration of the monomeric triangles?

We typically consider the triangle-to-lipid ratio in our assays. As shown in our new Supplementary Figure 12 (see also our response to Comment 1-10), the apparent depletion of monomeric triangles does not (solely) depend on the magnesium concentration. The assay comparing passivated and non-passivated triangles (Supplementary Figure 15, formerly 12) intentionally contained fewer vesicles than needed for complete sequestration of triangles by the vesicles. The disappearance of monomeric triangles at 65 mM is therefore just a consequence of these triangles assembling into shells. We have clarified this in the updated figure caption:

SF15, figure caption:

(...) GV membranes were added at half the ideal amount for DCV formation to retain a fraction of free monomeric triangles. (...)

To further explore the effect of triangle concentration on DCV yields, we performed additional experiments using SUVs, LVs, and GV membranes, in which we varied the triangle-to-lipid ratio. We observed distinct optima in DCV formation, which differed depending on vesicle size and swelling buffer used to prepare the vesicles. We attribute these differences to variations in lamellarity and, consequently, the accessible membrane area, and added the data into a new Supplementary Figure 14.

Lines 190-196:

DCV formation efficiency is also affected by the ratio of triangles to vesicles, which depends on vesicle size and swelling conditions (Supplementary Figure 14). Small vesicles (SVs) and LVs produced by extruding GV membranes showed a lower lipid optimum than untreated GV membranes, likely due to fewer multilamellar vesicles²⁶. Similarly, budding from GV membranes formed in salt-free sucrose buffer showed a lower Mg^{2+} optimum than from those swollen in caesium buffer, consistent with increased vesicle aggregation due to bivalent cations²⁷. In both cases, the accessible membrane area appears to be the key factor. (...)

Supplementary Figure 14 | DCV budding as a function of lipid quantity and vesicle size. **a**, Normalised DCV yields (lipid band intensities of octahedral and icosahedral DCVs, combined) from vesicles of varying sizes, quantified via agarose gel electrophoresis ($n = 3$). Budding reactions were prepared by adding the indicated quantities of lipids in the form of vesicles, ranging from SVs (\varnothing 50 nm), LVs (\varnothing 200 & 400 nm), to GVs, with a constant amount of triangles ($4.5 \mu\text{l}$ at 15 nM). DCVs are formed most efficiently between 100-200 pmol lipids for all tested vesicle sizes except GVs (note the broader range of the x-axis for GVs). This deviation may result from lamellarity differences, as multilamellar vesicles contribute less accessible membrane area per pmol lipid. Extruding GVs to produce smaller vesicles likely reduced their lamellarity²⁵. Data points were obtained from independent experiments using freshly prepared vesicles; the curves indicate their mean. **b**, Agarose gels probing DCV yields at varying amounts of lipids. Left: Exemplary gel of budding reactions using caesium buffer-filled GVs used to obtain the curves in a and b. Right: Budding from GVs filled with sucrose buffer, showing a lower lipid optimum. The absence of salts in sucrose buffer may yield fewer multilamellar vesicles, increasing the accessible membrane area per pmol lipid. Yellow-marked lanes indicate optima. Bottom: Cholesterol placement on the triangle surface.

Comment 1-12

The gel image in Supplementary Figure 13 appears to be cropped as the monomeric and oligomeric triangle bands are not visible. Could the authors include the full gel image? Additionally, the authors state “We finally obtained stronger lipid signals in samples initially incubated at 5 mM”, is there a reason for using the word ‘finally’ here? Could the authors show what happens when the concentration of Mg is lowered after the formation of DCVs? Does the shell fall apart leaving the vesicles or does the buried cholesterol keep the origami shell integrated?

Supplementary Figure 16 (formerly 13) shows the gel scanned in the lipid channel, where we generally only get bands for DCVs (i.e. assembled shells). We have now expanded the figure to also include the DNA channel, where monomers are faintly visible, though the majority is attached to the vesicles and thus stuck in the pockets.

Supplementary Figure 16 | Shell preassembly reduces DCV yields. (...)

Our usage of the word “finally” in this context refers to the step being the last in a sequence of steps (i.e. the final step).

To address the stability of DCVs upon magnesium reduction, we performed dialysis of the budding reaction into the original folding buffer (5 mM MgCl₂, 300 mM NaCl). As expected, in the absence of lipid vesicles, DNA shells disassembled into monomeric triangles under these low-salt conditions. In contrast, DCVs exhibited greater structural resilience. While some dissociated into monomers, others remained stably associated with vesicles. Notably, the remaining DCVs appeared morphologically altered (less ordered and more stressed compared to those formed and maintained under native (high Mg²⁺) conditions).

Lines 188-189:

(...) DCVs also display moderately increased resistance to low-salt conditions, whereas free shells fall apart into monomers (Supplementary Figure 13).

Supplementary Figure 13 | Stability of DCVs under low-salt conditions. To assess the structural stability of DCVs in reduced ionic strength, both free triangle assemblies (without vesicles) and budding reactions were dialysed against sodium buffer, lowering the MgCl₂ concentration from 60 mM to 5 mM. **a**, Stability of free shells. Top: Agarose gel electrophoresis (EtBr-stained) showing DNA shells before and after dialysis. Under high-salt conditions, triangles efficiently assembled into icosahedral shells. Upon Mg²⁺ depletion, the shells disassembled into monomeric triangles. Bottom: TEM micrographs corroborate the gel data, showing intact shells before dialysis and monomers afterwards. **b**, Stability of DCVs. Top: Agarose gel showing DCVs before and after dialysis (left: DNA channel (EtBr); right: lipid channel (DOPE-Atto643)). While DCVs persist post-dialysis, a laddering pattern beneath the octahedral DCV band in the DNA channel suggests partial destabilisation. Bottom: TEM images confirm the co-existence of intact DCVs and monomeric triangles post-dialysis. **c**, Close-up TEM micrographs of dialysed DCVs. Free triangles tend to cluster and appear associated with lipid patches. DCVs remain present but appear structurally compromised and less ordered compared to those kept at assembly conditions. All scale bars: 100 nm.

Comment 1-13 (minor)

To help grow the origami community and contribute to open research, I suggest the authors submit their origami designs to nanobase.org.

We are happy to share our design and will upload as the manuscript review process progresses going forward.

Comment 1-14 (minor)

In Fig 2d, the authors talk about the influence of orthogonal chol-oligos. Please provide an illustration indicating the location of these orthogonal chol-oligos (similar to Fig. 4a). Are these origamis with chol-oligos? Or are they just excess origami-free chol-oligos which are used in the bivesicular system? Could the authors clarify this?

The orthogonal chol-oligos are not complementary to any linker handles on the triangle and are instead added as free-floating oligos. We clarified this in the manuscript.

Lines 258-261:

(...) Remarkably, this approach still resulted in DCV formation and adding excess chol-oligos with an orthogonal sequence not complementary to the triangle linker handles did not improve yields (Fig. 3d & Supplementary Figure 20). (...)

Comment 1-15 (minor)

As a follow-up, in Supplementary Figure 4, which positions were chosen for placement of 1 and 3 chol-oligos (only the 9 chol-oligo illustration was shown in Supplementary Figure 3)? Could the authors provide some illustrations for the position of chol-oligos in Supplementary Figure 4?

We have added illustrations to SF4 and also other Supplementary Figures.

Comment 1-16 (minor)

The authors state that “spontaneous membrane scission outcompetes the completion of the DNA origami shell leading to the formation of DCVs” which in some cases leaves a residual “scar”. Could the authors comment on what prevents the excess monomeric triangles from completing the shell after budding? Could this relate to the reduced intensity of monomeric and oligomeric origami bands at 65 mM Mg for the non-passivating variant (refer to major suggestions)? How many instances of “scars” did the authors observe in their data? Please include some statistics.

We have added a discussion on DCV scars to the main text.

Lines 420-426:

The scars observed on some DCVs point towards steric hindrance at the bud neck, preventing shell closure. While these defects could, in principle, be repaired by incorporating free triangles, the efficiency of this depends on both their availability and their assembly state. Here, the assembly of free triangles at elevated Mg²⁺ concentrations may result in oligomers that are too large to fit into the scars of most DCVs. However, the rapid kinetics of DCV budding may outpace free triangle oligomerisation, allowing their incorporation into the DCVs and explaining why only a subset exhibits scars.

Scars are not always easily visible by ns-TEM due to particle orientation. However, more pronounced cases — such as half-shells or pentagonal DCVs — are easier to identify. We believe these pentagonal DCVs adopt an icosahedral base geometry, but are missing a large part of the DNA shell, resulting in their distinct pentagonal appearance. The most extreme case involves vesicles bound to a five-triangle cap (see Fig. 3a (top) for a schematic). We now provide statistics and details on these particles in Fig. 2a.

Comment 1-17 (minor)

The term “vesicle-coated DNA shells” seems somewhat confusing. I think DNA origami-coated vesicle is apt since individual origami structures assemble into a coating/shell. However, the vesicles do not coat/assemble on the surface of origami but rather the origami shell formation leads to their engulfment/invagination/encapsulation by vesicles. Could the authors comment on this and make the naming more intuitive?

We appreciate the opportunity to clarify our terminology. Our choice of the term *vesicle-coated DNA shells* (VCDs) is intended to typographically reflect the reversed topology compared to the outward budding process that produces DNA-coated vesicles (DCVs). In VCDs, the vesicle envelops the DNA shell, inverting the spatial relationship between membrane and scaffold.

While the vesicle does not actively “coat” the DNA in an assembly sense, it forms the outermost visible layer. We use the term “coated” here in a descriptive rather than mechanistic way, to mirror the naming of DCVs and to emphasise the symmetry between the two processes. We have therefore opted not to rename the particles.

Comment 1-18 (minor)

The manuscript mentions “... solution at 37 °C for up to multiple days”. Could the authors specify the duration more specifically or provide a range of days as they do in the methods section?

We have specified the duration as suggested.

Lines 127-129:

We then triggered the assembly of membrane-bound triangles by increasing the MgCl₂ concentration to 60-65 mM and incubating the solution at 37 °C for up to 3 d.

Comment 1-19 (minor)

Please indicate which channels were used for imaging for all gels (missing in some gel images like Supplementary Figures 3 and 4).

We have added the missing imaging channel information to all relevant gel images in the supplementary material, as well as to the corresponding figure captions in the main text.

Comment 1-20 (minor)

Please denote 'BD' in the figure legend for the distal positioning of cholesterol closer to the centre of the structure in Supplementary Figure 3. Additionally, please indicate how many chol-oligos were used in Supplementary Figure 3c. Finally, please use the word 'chol-oligos' instead of 'chol' in the gel images as that has been the terminology used throughout the manuscript (in Supplementary Figure 3 and for other figures as well).

We have updated the caption of Supplementary Figure 3 to list all variants in order of appearance and added the number of cholesterol used in panel c.

Supplementary Figure 3 (...) When cholesterol is positioned closer to the outer edge and the stacking contacts in a distal configuration (A_D), bands in an agarose gel appear shifted and smeared compared to the same triangles hybridised to unmodified (chol-free) oligos. This effect could be reduced by changing the orientation of the cholesterol to a proximal configuration with respect to the origami, such that the reach of the cholesterol is minimised (A_P). Chol-mediated interactions could be reduced further by positioning the linker handles closer to the centre of the structure, and thus further away from the stacking contacts (B_D & B_P). (...)

Regarding terminology, we use “chol-oligos” in the main text when referring to the full construct, but prefer the shorter “chol” in figure panels where the emphasis is on cholesterol as a membrane anchor. This avoids implying that free chol-oligos were added to the sample mixture when they were, in fact, pre-hybridised to the origami. For clarity, we have updated the labelling from “chol” to “chol-oligos” in Figure 3d and its channel-separated version in the supplement, where chol-oligos were indeed directly added to the mixture.

Fig. 3 (cropped)

Comment 1-21 (minor)

Although the reader can interpret upon careful reading, could the authors please clarify that the origami signal in the gel pocket of Supplementary Figure 11 (and some other images as well) might represent the GV bound triangles in monomeric/oligomeric form which were unable to form the DCVs and so doesn't enter the pocket? Additionally, please clarify that the monomeric and oligomeric origami signal in the Atto643 channel could correspond to the excess origami which might not be bound to GVs.

We have moved the data from the former Supplementary Figure 11 into main Figure 2b and added the clarification to the image caption and to the main text.

Fig. 2b, figure caption:

(...) DNA retained in the gel pocket reflects vesicle-bound triangles; the presence of a monomer band indicates saturated membranes. (...)

Lines 180-181:

(...) Membrane-bound triangles that failed to form DCVs the gel pockets.

Comment 1-22 (minor)

In Supplementary Figures 16 and 18, it appears as though the gels were run at higher magnesium concentration, as one could see a band corresponding to octahedral in the monomer lane. Please clarify this and ensure that all figure legends should include the gel conditions wherever they deviate from standard running conditions.

The gels in question (now in SF21 & SF24) were run at the same conditions as stated in the methods section (20 mM MgCl₂). These gel conditions preserve the shell structure whilst keeping triangles mostly in their monomeric state. Slightly higher concentrations (e.g. 22.5 mM) result in significant assembly of monomeric triangles in the gel, whilst lower concentrations could damage the shells. Nevertheless, we do occasionally observe low-grade assembly of the monomer control at 20 mM MgCl₂ which we attribute to small variations in the gel buffer, loading conditions (e.g. the time between loading and starting the gel, loading dye etc.) or origami structure (linker handle position, sequence etc.). Once assembly conditions are met, assembly occurs rapidly (shown in Fig. 3g). However, as the purpose of the monomer control is solely to indicate the running height of monomeric triangles, we do not consider low-grade assembly problematic. We have added a clarifying sentence to the figure captions.

SF21 & SF24, figure captions:

(...) Low-grade assembly of the monomer ctrl is caused by the gel running conditions at 20 mM MgCl₂. (...)

Reviewer 2

This manuscript reports the generation of budding vesicles with self-assembling DNA triangles modified with membrane anchors. Biomimetic DNA origami structures, mostly built to model the activities of BAR domains, dynamin, ESCRT and clathrin, have been reported to deform lipid membranes. Compared with previous studies, this work stands out because it (1) achieved well-defined self-assembling-dependent vesicle budding, (2) thoroughly investigated the design and experimental parameters that influences the deformation outcome, and (3) generated topologically controlled inward and outward-budding events. The membrane budding products and intermediates were visualized by negative-stain EM and cryoEM, the yield of the budding vesicles were estimated by gel electrophoresis. The high-quality data clearly support curvature induction by the self-assembling DNA triangles. The manuscript is well written, and I read it with great interest. Overall, this work highlights the programmability of the DNA-origami-based membrane-deforming platform and is a valuable addition to the high-precision membrane engineering toolbox.

Below are a few questions and suggestions:

Thank you for taking the time to review our manuscript and for your constructive feedback. We appreciate your kind remarks regarding the quality of our work, and your careful reading of the manuscript. Your comments on the need for additional controls and your suggestion to look deeper into the binding behaviour of our DNA nanostructures to phase-separated vesicles were particularly insightful, and we have worked to address each of your points in detail. We believe that your suggestions have contributed significantly to enhancing the quality and transparency of our study.

Comment 2-1

The authors investigated the effect of cholesterol number and positioning on the membrane deformation outcome. However, the copy number and positioning of cholesterol on DNA triangle are not entirely clear for each experiment (e.g., Supplementary Fig 4, 5, 6). Please clarify this for each piece of relevant data presented.

Thank you for your suggestion, we have now added schemes of the triangle variant used in each experiment to all relevant supplementary figures.

Comment 2-2

Budding happens more readily on low-tension membranes, as shown in Fig. 2c. The question is whether the budding under such conditions (i.e. hypertonic) still depends on the self-assembly of DNA shells (i.e., high Mg²⁺ and shape-complementary interfaces)?

We have tested this by comparing membrane-tethered triangles in isotonic and hypertonic (+700 mM Glycine) buffers and using both non-passivated and passivated triangles. We did not find any signs of directed budding as we do for DCV budding at assembly conditions. We included our findings into the manuscript and supplementary data.

Lines 240-243:

(...) When magnesium levels were not increased to trigger assembly, we also did not observe directed budding as for DCVs in both isotonic and hypertonic buffers, further confirming the necessity for assembly (Supplementary Figure 19). (...)

Supplementary Figure 19 | Effects of hypertonic conditions on membrane-bound triangles. a, Agarose gel of GV-bound triangles under isotonic (5 mM MgCl₂, 300 mM NaCl) or hypertonic (700 mM glycine, 31 mM NaCl, 5 mM MgCl₂) conditions. Both assembly-competent (left) and passivated triangles (middle) were tested

and compared to a vesicle-free assembly control (right). Under hypertonic conditions, increased triangle signal in the gel and a stronger monomer band suggest reduced membrane coupling, possibly due to glycine interfering with chol-oligo insertion or sequestration by micelles. DNA-lipid hybrids appear as smears or bands in the lipid channel. **b**, TEM of the same samples. At isotonic conditions, triangles often bind to micelles, small vesicles, or membrane patches in a disordered fashion, without forming shells. At hypertonic conditions, triangles are mostly monomeric and membrane-detached, consistent with weakened chol-mediated binding. Budding under hypertonic conditions as discussed in Fig. 3c & Supplementary Figure 18 may thus partially rely on both avidity and electrostatic attraction between vesicles and DNA triangles to strengthen membrane-tethering in presence of glycine. Although this data suggests the possibility of lipid extraction by a non-budding mechanism, an artefactual origin appears equally likely. Scale bars: 100 nm.

Comment 2-3

It is remarkable that in certain EM images, the DNA triangles almost exclusively localizes to the membrane buds, suggesting that complete membrane coverage by DNA structures may not be required when DNA triangles assemble into cages. The authors cited ref 14 to argue that attaching DNA origami structures to lipid vesicles can deform vesicles without requiring subunit assembly. However, this study (and many others such as ref 16) show that such deformation requires high surface coverage. It would be nice if the authors can elaborate on this point in the discussion.

This is a very interesting point, indeed! We have expanded the discussion to address it.

Lines 393-400:

Previous studies have shown that high surface coverage of membrane-bound DNA origami is often required for vesicle deformation^{14,16}. In contrast, our system induces budding even at low surface densities (Fig. 1d), likely due to the cooperative assembly of DNA triangles. This modular process allows local enrichment and gradual membrane bending driven by the free energy of assembly without the need for global coverage. Indeed, DCV formation occurs across a wide range of origami-to-vesicle ratios (Supplementary Figure 14), indicating that membrane remodelling is governed more by local assembly dynamics and diffusion-driven encounters than on overall density. (...)

Comment 2-4

Membrane binding DNA triangles modified by either cholesterol or biotin-neutravidin can similarly generate DNA-coated vesicles. But are they as efficient as each other? How does the yield compare to one another when all else (osmolarity, surface coverage, membrane-anchor positioning, etc.) are equal?

Supplementary Figure 20 shows samples using both anchor types under otherwise identical conditions. The two gel sections shown are parts of the same gel and thus allow for direct comparison. We have expanded the figure caption to stress this, and also discuss the comparison of yields in the main text.

Lines 261-263:

(...) Compared to DCVs obtained using chol, the distribution between icosahedral and octahedral DCVs is more balanced when using b-NAv, though overall DCV yields appear to be slightly lower. (...)

SF20, figure caption:

(...) Comparison of shell band intensities in the lipid channel indicates that cholesterol-anchored triangles produce slightly higher overall DCV yields, while b-NAv anchoring results in a greater proportion of icosahedral species. Notably, the monomer band is much fainter in the chol-mediated reaction compared to b-NAv, suggesting lower vesicle coverage — or alternatively, less aggregation — in the b-NAv samples under otherwise identical conditions.

Comment 2-5

Cholesterol-labeled DNA structures have been shown to partition in different domains of phase-separated lipid bilayers under certain conditions (e.g., Kanwa 2023 10.1002/admi . 202202500). Did the authors observe such phenomena? Did these membrane domains deform similarly or differently?

Thank you for this interesting suggestion! We were able to reproduce the key findings reported in the referenced study and have expanded both the main manuscript and the Supplementary Information accordingly to incorporate this additional data and contextualise it within our work.

Lines 298-300:

Consistent with previous reports³², we also observed preferential binding of DNA nanostructures to the Id-phase at low magnesium concentrations, with a shift towards the lo-phase upon triggering assembly (Supplementary Figure 23).

Supplementary Figure 23 | Membrane tethering of chol-functionalised triangles on phase-separated GVs as a function of MgCl₂ concentration. **a**, Fluorescence microscopy images of GVs with phase-separated membranes (red: Id-phase, LissRhod-PE; cyan: lo-phase, TF-chol) and DNA triangles (white, Atto643). At low MgCl₂ (5 mM), triangles preferentially localise to the Id-phase, consistent with previous reports³². **b**, At assembly conditions (65 mM MgCl₂), triangles increasingly localise to the lo-phase. Not all vesicles showed this reversal, possibly due to high triangle density hindering phase separation. For improved triangle visibility, the sample in b contained 5× more triangles than in a; however, the Id-phase bias at low MgCl₂ was also observed at higher triangle concentrations (data not shown).

Comment 2-6

In theory, the membrane budding process can be halted or even reversed by lowering the Mg²⁺ concentration, which stops and disrupts DNA shell assembly. Have the authors attempted this? I ask out of curiosity and would not regard the work as incomplete if this piece is missing.

Thank you for your interest! While we have not attempted to actively halt or reverse budding by lowering the Mg²⁺ concentration during the assembly process—primarily due to the rapid budding kinetics, which would always result in some degree of DCV formation—we have explored the effect of reducing Mg²⁺ concentration after budding has occurred. Interestingly, we found that membrane-bound DCVs exhibit moderately enhanced stability compared to free shells under these conditions. We have compiled these observations into the newly added Supplementary Figure 13.

Reviewer 3

Pinner et al. presented a DNA origami assembly system on lipid vesicles to mimic the virus assembly and control directional membrane budding. By leveraging the programmability of DNA origami triangles, this approach enables the formation of DNA-shell-coated vesicles, vesicle-coated DNA shells, and multivesicular DNA shells. The resulting DNA origami-based membrane budding system in some manner replicates key aspects of natural endocytic and exocytic pathways. Overall, the study presents clear and compelling results, making it highly relevant to researchers in DNA nanotechnology and synthetic biology. Especially, the presented structures are of exceptional quality, as always from Dietz's lab.

Thank you very much for your thorough and constructive review of our manuscript. We greatly appreciate the time and care you invested in evaluating our work. We took up your suggestion to quantify membrane curvatures and prepared a figure which we believe will help convincing readers of the membrane-deforming properties of membrane-confined triangle assembly. We are also grateful for making us aware of general flaws in the manuscript that we overlooked, such as redundancies in acronym definitions or ambiguous wording. Your thoughtful comments have helped us to present our findings more clearly and robustly, and we sincerely thank you for your insightful contributions.

Comment 3-1

While the authors have previously reported the assembly of DNA origami triangles into icosahedral shells (Reference 21), and the impact of cholesterol decoration on individual origami triangles and their assembly has been demonstrated via agarose gel electrophoresis (Figure S3), this evidence alone is insufficient to substantiate the claim in Line 78 that “and then validated their assembly into the expected icosahedral shells in the absence of membranes”. Additional TEM images would help validate this statement. Furthermore, for the discussions related to Figures S4-S6, additional TEM characterization is recommended to complement the agarose gel data and strengthen the analysis.

To support our claims, we have added representative TEM images to SF3, SF4 and SF5. For Supplementary Figure 6, the assembly reactions of cholesterol-functionalized triangles follow the same conditions as those in Supplementary Figure 5 (leftmost panels). Given that the gel shifts between samples with and without GVs are nearly identical across both figures, we have opted not to include additional TEM images in Supplementary Figure 6 to avoid redundancy.

Comment 3-2

For the key TEM images that support the main results of this study (Figure 1d, Figure 3b and Figure 4b), only some cropped TEM images are presented. It is necessary to include corresponding overview TEM images to estimate the yield of this artificial membrane budding system.

Figure 1d shows vesicle deformation in response to origami assembly, whereas Figures 4b (formerly 3b) and 5b (formerly 4b) show the budding products. We thus assumed that you meant to refer to Figure 1c, which also shows DCVs as budding products. We have added larger TEM images into SF11 (for DCVs (Fig. 1d)), SF25 (for VCDs (Fig. 4b)), and SF26 (for VCDCVs (Fig 5b)).

Supplementary Figure 11 | DCVs obtained from triangles with 3, 6 or 9 cholesterol. Full fields of view from negatively stained TEM micrographs of DCV samples formed using triangles with **a**, 3; **b**, 6; or **c**, 9 cholesterol. Irregular particles (e.g., unusually large or ambiguous shapes) and free monomers or oligomers were not marked. All images were acquired under identical conditions. Scale bars: 100 nm. **d**, Frequency of particle subspecies by cholesterol count. Besides octahedral and icosahedral DCVs, pentagonal DCVs—likely representing scarred icosahedra—were consistently observed. Lower cholesterol content resulted in more empty shells, while higher counts promoted the formation of octahedral DCVs. Triangles with 3 cholesterol also produced a distinct population of incomplete DCVs with small internal vesicles, less prevalent at higher cholesterol numbers. Although some may be staining artefacts, their enrichment in low-cholesterol samples suggests weaker membrane binding under these conditions. Particles with ambiguous or unclear morphology were excluded from the analysis. Numbers in brackets indicate absolute particle counts.

Supplementary Figure 25 | CryoEM images of VCDs, full fields-of-view. Scale bars: 100 nm.

Supplementary Figure 26 | CryoEM images of VCDCVs, full fields-of-view. VCDCVs were produced by adding concentrated DCVs to LVs covered with chol-oligos integrated into them. Unlike VCD budding reactions, in which triangles are first bound to vesicles in a monomeric state, VCDCV budding requires mixing of DCVs and LVs at high magnesium conditions (60 mM) to ensure shell stability. These conditions promoted budding of DCVs into the freshly added, oligo-coated LVs, but also caused aggregation mediated by electrostatic interactions, DCVs binding to two LVs simultaneously, and pairing of residual single-stranded linker handles on the triangles, lowering the yield of VCDCVs. Optimisation of linker handles on the triangle surface and improving the DCV-LV coupling strategy may improve yields. Scale bars: 100 nm.

Comment 3-3

Regarding the schematic in Figure 1a, the membrane appears to be depicted as a 2D surface. However, all membrane budding experiments described in the manuscript were conducted on 3D lipid vesicle membranes. Therefore, it is recommended to revise Figure 1a to represent a 3D membrane instead of a 2D one.

Figure 1a is intended as a conceptual illustration of the budding process on a local segment of a giant vesicle. Given the large size difference between GVs (10–20 μm) and DNA origami triangles ($\sim 40\text{ nm}$), the local membrane curvature is negligible from the perspective of the assembling structures. We therefore opted for a simplified 2D depiction to maintain clarity and focus. To address the reviewer’s concern, we have added the following clarification to the figure caption:

Fig. 1, figure caption:

(...) A flat, rectangular segment is shown to represent a local portion of a giant vesicle (GV, purple). (...)

Comment 3-4

In the first section of “DNA-shell-coated vesicles form by membrane budding,” the conclusion in Line 381 that “curvature increases gradually and closely follows the intrinsic curvature of the assembling shells” and the hypothesis in Line 138 that “Considering that the increasingly high curvature at the bud neck can become a steric barrier to completion of the DNA origami shell, we believe the occurrence of scars to be a signature of the proposed budding process” are drawn primarily from the TEM images in Figures 1d and 2b. However, these images alone provide limited evidence to fully support these claims. To substantiate the statement that “curvature increases gradually and closely” and the hypothesis regarding “the increasingly high curvature at the bud neck,” real-time monitoring of the 2D membrane using AFM would be a valuable approach. Alternatively, a statistical analysis of curvature based on similar TEM features observed at different stages, as shown in Figure 3c, could also strengthen the argument.

Thank you for your suggestion. We have measured the curvatures of 27 membrane buds and compiled our findings into Supplementary Figure 10. We performed a curvature analysis of 27 individual membrane buds captured by cryoEM and compiled the results in the new Supplementary Figure 10. The data show a range of curvatures, supporting a gradual progression from shallow to more pronounced membrane bending during bud maturation.

Supplementary Figure 10 | Curvature analysis of growing buds. CryoEM micrographs of outward-directed buds on LVs, sorted by decreasing curvature from top to bottom. Larger triangle assemblies adopting icosahedral geometry (e.g., bud 8) induce shallower curvature near completion than similarly mature octahedral assemblies (e.g., buds 1 & 2). In contrast, early-stage assemblies, such as the triangle dimer shown in bud 27, induce only a small bump in the vesicle membrane, illustrating the coupling between curvature maturation and shell completion. Most buds shown lie in between both extremes and differ only slightly in their curvature values. Curvatures (κ , in μm^{-1}) were measured by curve fitting using Kappa (Fiji) and

correspond to the inverse radius of the osculating circle approximating the membrane curvature underneath the triangle coat. As buds may be viewed at an angle, measured values may underestimate true curvature. Scale bar: 100 nm (all images).

Comment 3-5

Is it possible that the DNA origami triangles first assemble in bulk before being loaded onto vesicles? Long-term AFM monitoring on a 2D membrane may also provide insights into this process.

As shown in Supplementary Figure 16, DCV yield is markedly reduced when GVs are added to a reaction 4 hours after assembly initiation, compared to a control where GVs and triangles are mixed simultaneously. Preassembled samples predominantly formed icosahedral shells, whereas the control produced a greater fraction of octahedral structures—consistent with vesicle-free versus membrane-associated assembly, respectively (see also Supplementary Figure 6). These findings support the conclusion that DCV formation primarily relies on membrane-templated assembly. However, the presence of DCVs in preassembled samples suggests that small intermediates (e.g., dimers) may still bind to vesicles at later stages. We have expanded the figure caption to clarify our findings.

Supplementary Figure 16 | Shell preassembly reduces DCV yields. To assess how preassembled structures affect DCV formation, cholesterol-functionalized origami triangles were preincubated without GVs for 4 h at 37 °C in either low (5 mM) or high (60 mM) MgCl_2 conditions—promoting monomeric or assembled states, respectively. Following this, GVs containing a small fraction of DOPE-Atto643 were added. In the 5 mM condition, MgCl_2 was then adjusted to 60 mM to allow DCV formation. Samples were analysed by agarose gel electrophoresis at two timepoints: immediately after GV addition ($t=0$ d), and after 2.5 d of incubation at 37 °C. At $t=0$ d, both conditions showed similarly faint lipid band intensities, indicating comparable vesicle binding. However, after incubation, the 5 mM preincubation condition produced noticeably stronger lipid signals, suggesting higher DCV yields. High- Mg^{2+} preincubation favoured the formation of icosahedra, while the low- Mg^{2+} condition supported greater formation of octahedra upon membrane addition. The icosahedral band in the DNA channel of the sample preincubated at 60 mM Mg^{2+} does not translate into the lipid channel, indicating that these are empty shells and not DCVs. These observations support the idea that membrane-bound assembly alters the structural dynamics (see also Supplementary Figure 6) and is key for efficient DCV production. Reduced yields in the preassembled samples likely result from steric constraints that hinder binding of larger intermediates to vesicles or prevent the induction of sufficient curvature. Nonetheless, the detection of DCVs even in these conditions implies that small oligomers—such as dimers—can still engage with membranes and complete shell formation. Larger assemblies may also bind but fail to mature into buds due to limited curvature or monomer availability.

Comment 3-6

In Figure 1b and Figure S7, the authors report that “At low-salt conditions, triangles adsorb onto lipid vesicles even if not hybridised to chol-oligos. By adding 300 mM NaCl, non-specific origami-vesicle association is suppressed without interfering with cholesterol-mediated association.” However, the exact NaCl concentration in these “low-salt conditions” is unclear. Based on the figure caption of Figure S7, it appears to be zero. Additionally, the rationale behind selecting a NaCl concentration of 26.3 mM in Supplementary Table 25 and 5 mM in the imaging buffer and Isotonic assembly buffer in Supplementary Table 26 requires clarification. Given that these values are significantly lower than 300 mM, it is important to explain how non-specific binding is mitigated under such low-salt conditions. These aspects should be addressed in both the manuscript and the Supplementary Information.

The “low-salt” condition in Supplementary Figure 7 corresponds to imaging buffer B, as detailed in the Methods section; we have now added the exact salt composition to the figure caption for clarity.

SF7, figure caption:

(...) At low-salt conditions (5 mM MgCl₂, 5 mM NaCl), triangles adsorb onto lipid vesicles even if not hybridised to chol-oligos. By adding additional 300 mM NaCl, non-specific origami-vesicle association is suppressed without interfering with cholesterol-mediated association. (...)

Regarding the inclusion of low NaCl concentrations in the other buffers: We base our buffers off TE buffer with 5 mM NaCl and add other additives, such as MgCl₂ or sugar, as required. Because of this, most of our buffers contain a default of 5 mM NaCl. The conditions listed in Supplementary Table 25 state the final composition of the samples in our study. The NaCl concentration in these samples is the result of GVs (resuspended in sodium buffer (300 mM NaCl)) with other components not containing NaCl. Our setup required alterations to be able to span a broad range of tonicities. As a result, we had to accept low NaCl concentrations that may no longer prevent non-specific adsorption. To rule out any unexpected effects, we included a budding control sample in our assay prepared as a standard budding reaction at near-isotonic conditions. In this sample, triangles were coincubated with GVs at 300 mM NaCl before triggering assembly. Despite this, the isotonic control had similar DCV yields as the +12% sample. While we avoid drawing conclusions from the direct comparison of budding efficiencies between the glycine-free control and the samples in the tonicity screen, the overall band intensities do not suggest significant (if any) interference of non-specific adsorption.

We have updated Supplementary Table 25 to include this isotonic control and added an explanatory note to the table caption.

Supplementary Table 25 | Tonicity screen sample compositions. Composition of the samples shown in Fig. 3c. The magnesium optimum for triangle assembly increased with increasing glycine concentrations. Concentrations above or below the optima presented in this table result in incomplete or impaired assembly. The isotonic control sample was conducted as a standard budding reaction, where triangles were tethered to the vesicles at high-salt conditions (5 mM MgCl₂, 300 mM NaCl). Addition of isotonic magnesium solution to trigger assembly lowered the overall NaCl concentration to 219.5 mM.

We have also added the following sentence to the main text to address this matter:

Lines 237-240:

Although the tonicity screen used lower NaCl concentrations than typically needed to prevent non-specific adsorption, DCV yields in the mildly hypertonic range were comparable to those of the control prepared at standard conditions, suggesting minimal interference with budding.

Comment 3-7

The effect of DNA origami triangle density or quantity on individual vesicles in relation to membrane budding should be experimentally verified.

We performed a set of experiments to investigate how different triangle-to-lipid ratios affect DCV formation, using small, large, and giant vesicles. These experiments revealed distinct lipid optima that varied depending on vesicle size and the buffer conditions used during swelling. We believe these differences are primarily due to variations in vesicle lamellarity, which affect the amount of membrane surface accessible for triangle binding and subsequent budding.

Lines 190-196:

DCV formation efficiency is also affected by the ratio of triangles to vesicles, which depends on vesicle size and swelling conditions (Supplementary Figure 14). Small vesicles (SVs) and LVs produced by extruding GVs showed a lower lipid optimum than untreated GVs, likely due to fewer multilamellar vesicles²⁶. Similarly, budding from GVs formed in salt-free sucrose buffer showed a lower Mg²⁺ optimum than from those swollen in caesium buffer, consistent with increased vesicle aggregation due to bivalent cations²⁷. In both cases, the accessible membrane area appears to be the key factor.

Supplementary Figure 14 | DCV budding as a function of lipid quantity and vesicle size. **a**, Normalised DCV yields (lipid band intensities of octahedral and icosahedral DCVs, combined) from vesicles of varying sizes, quantified via agarose gel electrophoresis ($n = 3$). Budding reactions were prepared by adding the indicated quantities of lipids in the form of vesicles, ranging from SVs (\varnothing 50 nm), LVs (\varnothing 200 & 400 nm), to GVs, with a constant amount of triangles ($4.5 \mu\text{l}$ at 15 nM). DCVs are formed most efficiently between 100-200 pmol lipids for all tested vesicle sizes except GVs (note the broader range of the x-axis for GVs). This deviation may result from lamellarity differences, as multilamellar vesicles contribute less accessible membrane area per pmol lipid. Extruding GVs to produce smaller vesicles likely reduced their lamellarity²⁵. Data points were obtained from independent experiments using freshly prepared vesicles; the curves indicate their mean. **b**, Agarose gels probing DCV yields at varying amounts of lipids. Left: Exemplary gel of budding reactions using caesium buffer-filled GVs used to obtain the curves in a and b. Right: Budding from GVs filled with sucrose buffer, showing a lower lipid optimum. The absence of salts in sucrose buffer may yield fewer multilamellar vesicles, increasing the accessible membrane area per pmol lipid. Yellow-marked lanes indicate optima. Bottom: Cholesterol placement on the triangle surface.

Comment 3-8

The authors summarize in Line 284: "In summary, we present a versatile platform for engineering vesicles of controlled size featuring addressable endo- or exoskeletons." However, given that the lipid vesicle sizes shown in Figure 1b vary over a broad range, it is also recommended to investigate and demonstrate the impact of vesicle size on membrane budding.

Please refer to our response to comment 3-7, where we also address the influence of different vesicle sizes on budding efficiency.

Comment 3-9 (minor)

In line 34, the results on DNA origami in this study do not fully support the statement: "We hypothesized that any material capable of self-assembly into curved shapes could act as a molecular scaffold for membrane budding." It is recommended to clarify this statement to better reflect the findings.

Our intention with that line was to convey the broader idea that nanoscale budding machinery does not have to be protein-based like clathrin-mediated endocytosis. We have revised the sentence to better reflect our focus on DNA origami as a non-protein scaffold capable of driving membrane budding.

Lines 33-36:

We hypothesised that also other materials capable of self-assembling into curved architectures beyond proteins could act as a molecular scaffold for membrane budding, with DNA origami emerging as a promising candidate.

Comment 3-10 (minor)

Typically, large-sized lipid vesicles are referred to as “giant unilamellar vesicles (GUVs),” as mentioned in Line 45. To maintain consistency, it is suggested to replace “giant vesicles (GVs)” and “large vesicles (LVs)” with “giant unilamellar vesicles (GUVs)” and “large unilamellar vesicles (LUVs)” throughout the manuscript.

While we are aware that the terms GUVs and LUVs are commonly used, we have intentionally chosen to use “GVs” and “LVs” to avoid suggesting that the vesicles in our study are exclusively unilamellar. Most standard preparation methods—including the one we use—typically produce vesicles with mixed lamellarity. To reflect this more accurately, we opted for more general terminology. We also cite reference 26, which specifically examines the lamellarity of large vesicles, to support this choice.

- 26. Scott, H. L. *et al.* On the Mechanism of Bilayer Separation by Extrusion, or Why Your LUVs Are Not Really Unilamellar. *Biophys. J.* **117**, 1381–1386 (2019).

Comment 3-11 (minor)

There are two instances of the abbreviation “(DCV)” in Lines 61 and 74. Additionally, the abbreviation “(VCD)” appears in Lines 61, 74, 219, and 418.

Thank you for pointing this out! We have revised the manuscript to define each abbreviation only at its first occurrence: “DNA-shell-coated vesicles (DCVs)” is now defined in line 82, and “vesicle-coated DNA shells (VCDs)” in line 324. All other redundant definitions have been removed for clarity and consistency.

Comment 3-12 (minor)

The experimental description should be more detailed. For example, in Line 99, the statement: “We then triggered the assembly of membrane-bound triangles by increasing the $MgCl_2$ concentration and incubating the solution at 37 °C for up to multiple days” lacks specificity. It would be clearer to specify the final Mg^{2+} concentration and replace “multiple days” with a precise duration (e.g., 48 or 72 hours).

We have specified the experimental procedures as requested.

Lines 127-129:

We then triggered the assembly of membrane-bound triangles by increasing the $MgCl_2$ concentration to 60-65 mM and incubating the solution at 37 °C for up to 3 d.

Line 797:

Reactions were incubated at 37 or 40 °C for ≥ 16 h, up to 3 d, unless stated otherwise.

Lines 912-914:

For inward budding, origami triangles (3 μ l at 20 nM) with shell-outward linker handles were mixed with LVs (1500 pmol lipids; approx. 200 nm diameter) and incubated at 37 °C for 1-3 d.

Comment 3-13 (minor)

The manuscript includes only two references published after 2023. It is suggested to update the references and incorporate discussions on more recent advancements in DNA nanotechnology and synthetic biology to ensure the work is aligned with the latest research.

We have included three more recent publications into our references, and updated the manuscript accordingly. Ref 21 displays another case of DNA origami-mediated vesicle deformation, and refs 37-38 offers insights into polymorphisms of DNA origami assemblies.

- 21. Fan, S. *et al.* Morphology remodelling and membrane channel formation in synthetic cells via reconfigurable DNA nanorrafts. *Nat. Mater.* **24**, 278–286 (2025).

- 37. Hayakawa, D. *et al.* Geometrically programmed self-limited assembly of tubules using DNA origami colloids. *Proc. Natl. Acad. Sci.* **119**, e2207902119 (2022).

- 38. Videbæk, T. E. *et al.* Economical routes to size-specific assembly of self-closing structures. *Sci. Adv.* **10**, eado5979 (2024).

Comment 1-1

Please include a scale bar in Supplementary Figure 17. Furthermore, please include the scale bar in the caption in Supplementary Figures 3 and 23. Also, please include the caption for Supplementary Figure 3d.

We thank the reviewer for making us aware of these errors. We added all missing scale bars and fixed the figure captions. To improve the manuscript structure, we moved our tomography data (formerly Fig. 3a & SF17) into the new Figure 2c. In the process, we removed SF17 from the Supplementary Information (the data is now fully included in Fig. 2c), so that SF23 is now SF22.

Supplementary Figure 3

(...) **d**, Representative TEM micrographs of the assemblies shown in c (chol-bearing triangles). Scale bars: 100 nm.

Supplementary Figure 22

(...) Scale bars: 10 μ m.

Comment 1-2

Are there additional representative images for Supplementary Figure 7?

We have added additional images to SF7.

Comment 1-3

Would the authors consider sorting buds 1-27 as increasing curvature from top to bottom in Supplementary Figure 10? In this way, the process of budding can be seen as a gradual increase in bud curvature and would be more intuitive.

We have updated SF10 to show increasing curvatures from top to bottom, as suggested.

Supplementary Figure 10 | Curvature analysis of growing buds. CryoEM micrographs of outward-directed buds on LVs, sorted by increasing curvature from top to bottom. Larger triangle assemblies adopting icosahedral geometry (e.g., bud 20) induce shallower curvature near completion than similarly mature octahedral assemblies (e.g., buds 26 & 27). In contrast, early-stage assemblies, such as the triangle dimer shown in bud 1, induce only a small bump in the vesicle membrane, illustrating the coupling between curvature maturation and shell completion. (...)

Comment 1-4

Just out of curiosity, is there a reason for redundant figures in the main (2a and 3a) and supplementary (SF 11d and 17a)?

As mentioned in our response to Comment 1-1, we have now shifted SF17 (and Fig. 3a) into Figure 2c.

As space is not an issue in supplementary figures, we opted to also include the data already presented in Fig. 2a into SF11d. We believe that this improves readability as readers interested in a particular aspect of our manuscript can find all available information in one place without having to switch back and forth between the main text and the supplement.